# CamI2V: Camera-Controlled Image-to-Video Diffusion Model

## Abstract

Recent advancements have integrated camera pose as a user-friendly and physics-informed condition in video diffusion models, enabling precise camera control. In this paper, we identify one of the key challenges as effectively modeling noisy cross-frame interactions to enhance geometry consistency and camera controllability. We innovatively associate the quality of a condition with its ability to reduce uncertainty and interpret noisy cross-frame features as a form of noisy condition. Recognizing that noisy conditions provide deterministic information while also introducing randomness and potential misguidance due to added noise, we propose applying epipolar attention to only aggregate features along corresponding epipolar lines, thereby accessing an optimal amount of noisy conditions. Additionally, we address scenarios where epipolar lines disappear, commonly caused by rapid camera movements, dynamic objects, or occlusions, ensuring robust performance in diverse environments. Furthermore, we develop a more robust and reproducible evaluation pipeline to address the inaccuracies and instabilities of existing camera control metrics. Our method achieves a 25.64% improvement in camera controllability on the RealEstate10K dataset without compromising dynamics or generation quality and demonstrates strong generalization to out-of-domain images. Training and inference require only 24GB and 12GB of memory, respectively, for 16-frame sequences at 256×256 resolution. We will release all checkpoints, along with training and evaluation code. Dynamic videos are available for viewing on our supplementary anonymous web page.

**(a) Explain the Principle of Condition:**

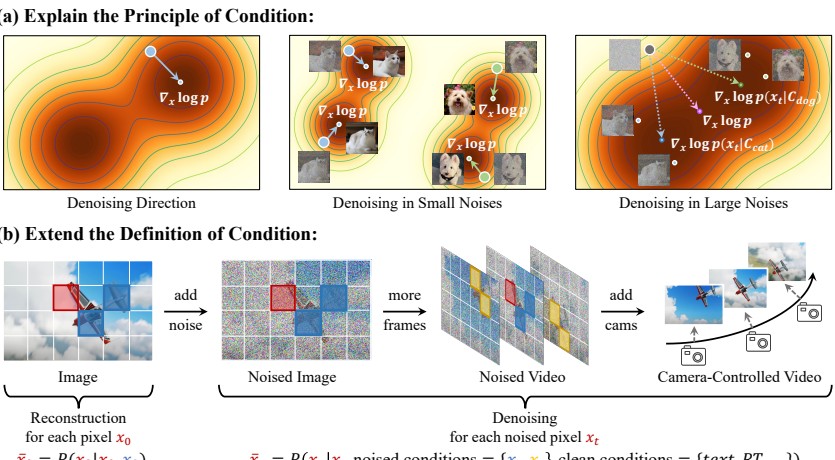

Denoising Direction · Denoising in Small Noises · Denoising in Large Noises

**(b) Extend the Definition of Condition:**

Image · Noised Image · Noised Video · Camera-Controlled Video

Reconstruction for each pixel $x_0$
$\bar{x}_0 = P(x_0|x_0, x_0)$

Denoising for each noised pixel $x_t$
$\bar{x}_0 = P(x_0|x_t, \text{noised conditions} = \{x_t, x_t\}, \text{clean conditions} = \{text, RT, \dots\})$

Figure 1: **Rethinking condition in diffusion models.** Diffusion models denoise along the gradient of log probability density function. At large noise levels, the high density region becomes the overlap of numerous noisy samples, resulting in visual blurriness. We point out that *the effectiveness of a condition depends on how much uncertainty it reduces*. From a new perspective, we categorize conditions into *clean conditions* (e.g. texts, camera extrinsics) that remain visible throughout the denoising process, and *noisy conditions* (e.g. noised pixels in the current and other frames) whose deterministic information $\alpha_t x_0$ will be gradually dominated by the randomness of noise $\sigma_t \epsilon$.

Figure 2: **Comparison of existing attention mechanisms for tracking displaced noised features.** Temporal attention is limited to features at the same location of picture, rendering it ineffective for significant camera movements. In contrast, 3D full attention facilitates cross-frame tracking due to its broad receptive field. However, high noise levels can obscure deterministic information, hindering consistent tracking. Our proposed epipolar attention aggregates features along the epipolar line, effectively modeling cross-frame relationships even under high noise conditions.

# 1 INTRODUCTION

The remarkable 3D consistency demonstrated in videos generated by Sora (Brooks et al., 2024) has highlighted the powerful capabilities of diffusion models (Ho et al., 2020; Rombach et al., 2022), showcasing their potential as a world simulator. Many researchers have attempted to enable the model to understand real-world knowledge (Chen et al., 2023a; Liu et al., 2023).

Condition or guidance (Ho & Salimans, 2022; Dhariwal & Nichol, 2021) is widely recognized as a crucial factor in enhancing generation quality. This is attributed to the fundamental principles that diffusion models denoise along the gradient of the log probability density function (score function) (Song et al., 2020), moving towards a high density region. However, this characteristic has varying effects at different noise levels (Tang et al., 2023a). As shown in Fig. 1(a), the high density region under high noise level becomes the overlap of numerous noisy samples, resulting in visual blurriness. By providing the model with conditions such as $c_{dog}$ and $c_{cat}$, it can rapidly eliminate incorrect generations. This illustrates that **adding more conditions can guide the model towards desired outcomes while reducing uncertainty.**

Consequently, *incorporating physics-related or more detailed conditions into the diffusion model is an effective way of improving its world understanding*. Considering that video generation requires providing condition for each frame, it is essential to identify a condition that is physics-related but also user-friendly. Recently, some camera-conditioned text-to-video diffusion models such as MotionCtrl (He et al., 2024a) and CameraCtrl (Wang et al., 2024d) have proposed using camera poses of each frame as a new type of condition. However, these methods simply inject camera conditions through a side input (like T2I-Adapter (Mou et al., 2024)) and neglect the inherent physical knowledge of camera pose, resulting in imprecise camera control, inconsistencies, and also poor interpretability.

In this paper, we identify one of the key challenges of camera-controlled image-to-video diffusion models as *how to effectively model noisy cross-frame interactions to enhance geometry consistency and camera controllability*. As illustrated in Fig. 2, separated spatial and temporal attention serves as an indirect form of 3D attention. The cross-frame interaction in temporal attention is confined to features at the same location in the image, rendering it ineffective for tracking significant movements resulting from large camera shifts. 3D full attention is widely applied in advanced video diffusion models such as OpenSora (Zheng et al., 2024) and CogVideoX (Yang et al., 2024b), due to its extensive receptive field. From the novel perspective of the noisy conditions mentioned in Fig. 1, the broad receptive field of 3D full attention allows it to access more noisy conditions. However, we argue that **accessing more noisy conditions does not necessarily reduce uncertainty** and thus not necessarily lead to better performance due to the randomness inherent in the noise. As previously highlighted in Fig. 1, **the quality of a condition is determined by its ability to reduce the model's uncertainty, rather than its quantity**.

To address these issues, we have found that **applying epipolar constraints is one of the most suitable way to prevent the model from being misled by noise**. By restricting attention to features along the epipolar lines, the model can interact with more relevant and less noisy information, improving cross-frame interactions in diffusion models. Specifically, we propose to apply Plücker coordinates (Plücker, 1828) as absolute 3D ray embedding for implicit learning of 3D space and propose a epipolar attention mechanism that introduces an explicit constraint. By doing so, our

approach minimizes the search space and reduces potential errors, ultimately enhancing 3D consistency across frames and improving overall controllability. Additionally, inspired by Timothée et al. (2024), we incorporate register tokens into epipolar attention to address scenarios where there are no intersections between frames, often caused by rapid camera movements, dynamic objects, or occlusions.

For inference, we propose a multiple classifier-free guidance scale to control images, text, and camera respectively. If needed, several forward passes can be combined into a single pass by absorbing the scales of image, text, and camera into the model input, similar to timestep conditioning according to (Meng et al., 2023). For evaluation, we identify inaccuracies and instability in the current measurements of camera controllability due to the intrinsic limitations of SfM-based methods such as COLMAP (Schonberger & Frahm, 2016), which rely on identifying keypoint pairs and is quite challenging on generated videos with low resolution, high frame stride, and 3D inconsistencies. Considering the importance of accurate evaluation in this field, we establish a more robust, precise, and reproducible evaluation pipeline by implementing several enhancements. More details are provided in Section 5.

We conduct experiments on the RealEstate10k dataset and evaluate video generation quality using FVD (Unterthiner et al., 2018), as well as camera controllability metrics including RotError, TranError (Wang et al., 2024d), and CamMC (He et al., 2024a). The results demonstrate that the proposed epipolar attention mechanism across all noised frames significantly enhances geometric consistency and improves camera controllability. To facilitate further research, we will release all models trained on open-source frameworks such as DynamiCrafter, along with high-resolution checkpoints and training/evaluation codes, as soon as possible. To summarize, our key contributions are as follows:

- We identify one of the key challenges of camera-controlled image-to-video diffusion models as effectively modeling noisy cross-frame interactions to enhance geometry consistency and camera controllability.

- Well-motivated by the relationship between the quality of a condition and its ability to reduce uncertainty, we innovatively interpret noisy cross-frame features as a form of noisy condition and propose to apply epipolar attention to access an optimal amount of noisy condition. We also address scenarios where epipolar lines disappear by register tokens.

- We point out and analyze the reasons for inaccurate measurement of camera controllability caused by the inherent limitations of SfM evaluator and re-establish a more robust, accurate and reproducible evaluation pipeline. We achieve a 32.96%, 25.64%, 20.77% improvement over CameraCtrl on RotErr, CamMC, TransErr on the RealEstate10K dataset without compromising dynamics, generation quality, or generalization on out-of-domain images.

## 2    RELATED WORK

**Diffusion-based Video Generation.**    With the advancement of diffusion models (Rombach et al., 2022; Ramesh et al., 2022; Zheng et al., 2022), video generation technology has progressed significantly. Given the scarcity of high-quality video-text datasets (Blattmann et al., 2023a;b), many researchers have sought to adapt existing text-to-image (T2I) models for text-to-video (T2V) generation. Some efforts involve integrating temporal blocks into original T2I models, training these additions to facilitate the conversion to T2V models. Examples include AnimateDiff (Guo et al., 2023), Align your Latents (Blattmann et al., 2023b), PYoCo (Ge et al., 2023), and Emu video (Girdhar et al., 2023). Additionally, methods such as LVDM (He et al., 2022), VideoCrafter (Chen et al., 2023a; 2024b), ModelScope (Wang et al., 2023a), LAVIE (Wang et al., 2023c), and VideoFactory Wang et al. (2024a) have adopted a similar structure, using T2I models as initialization weights and fine-tuning both spatial and temporal blocks to achieve better visual effects. Building on this foundation, Sora (Brooks et al., 2024) and CogVideoX (Yang et al., 2024b) have significantly enhanced video generation capabilities by introducing Transformer-based diffusion backbones (Peebles & Xie, 2023; Ma et al., 2024a; Yu et al., 2024) and leveraging 3D-VAE technology, thereby opening up the possibility of world simulators. Furthermore, works such as Dynamicrafter (Xing et al., 2023), SVD (Blattmann et al., 2023a), Seine (Chen et al., 2023b), I2vgen-XL (Zhang et al., 2023b), and PIA (Zhang et al., 2024) have extensively explored image-to-video generation, achieving substantial progress.

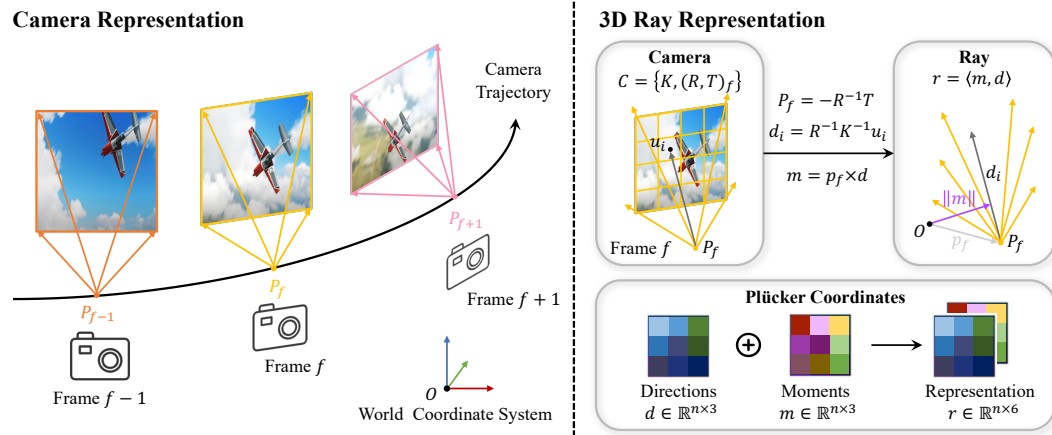

Figure 3: **Parameterizations for cameras.** Left: Camera representation and trajectory visualization in the world coordinate system. Right: The transformation from camera representations to 3D ray representations as Plücker coordinates given pixel coordinates.

**Controllable Generation.** With the development of image controllable generation technology (Zhang et al., 2023a; Jiang et al., 2024; Mou et al., 2024; Zheng et al., 2023; Peng et al., 2024; Ye et al., 2023; Wu et al., 2024b; Song et al., 2024; Wu et al., 2024d), video controllable generation has gradually become a highly focused direction. Significant progress has been made in areas such as pose (Ma et al., 2024b; Wang et al., 2023b; Hu, 2024; Xu et al., 2024b), trajectory (Yin et al., 2023; Chen et al., 2024a; Li et al., 2024; Wu et al., 2024a), subject (Chefer et al., 2024; Wang et al., 2024c; Wu et al., 2024c), and audio (Tang et al., 2023b; Tian et al., 2024; He et al., 2024b), greatly facilitating users to generate desired videos according to their needs.

**Camera-controlled Video Generation.** AnimateDiff (Guo et al., 2023) utilizes LoRA (Hu et al., 2021) fine-tuning to achieve specific camera movements. MotionMaster (Hu et al., 2024) and Peek-aboo (Jain et al., 2024) explore a training-free method for coarse-grained camera movement generation, but they lack precise control. VideoComposer (Wang et al., 2024b) offers global motion guidance by adjusting pixel-level motion vectors. In contrast, MotionCtrl (Wang et al., 2024d), CameraCtrl (He et al., 2024a), and Direct-a-Video (Yang et al., 2024a) incorporate camera pose information as side input; however, these methods primarily focus on text-to-video generation and do not effectively leverage 3D geometric priors in camera pose. CamCo (Xu et al., 2024a) also facilitates controllable camera generation in the image-to-video task by using epipolar attention (Kant et al., 2024; Tseng et al., 2023) to ensure consistency between generated frames and a single reference frame only. However, it does not account for scenarios where frames lack overlapping regions with the reference frame and can thus be regarded as a degenerate version of our approach.

## 3 METHOD

### 3.1 PRELIMINARIES

**3D Ray Embedding.** We follow CameraCtrl (He et al., 2024a) to apply plücker embedding as global positional embedding. Considering camera intrinsics $K \in \mathbb{R}^{3 \times 3}$ and extrinsics (rotation $R \in \mathrm{SO}(3)$, translation $T \in \mathbb{R}^3$), it parameterizes the transform from world coordinates to pixel coordinates by projection $u = K[R \mid T]x$. This low-dimensional representation may hinder neural networks from direct regression. Instead, we follow (Tseng et al., 2023) to represent cameras as ray bundles:

$$\mathcal{R} = \{r_1, \ldots, r_n\}, \tag{1}$$

where each ray $r_i \in \mathbb{R}^6$ is associated with a known pixel coordinate $u_i$. Each ray $r$ can be parameterized by ray direction $d \in \mathbb{R}^3$ from camera center $P \in \mathbb{R}^3$ as Plücker coordinates:

$$r = \langle m, d \rangle \in \mathbb{R}^6, \tag{2}$$

where $m = p \times d \in \mathbb{R}^3$ is the moment vector. When normalize $d$ to unit length, the norm of the moment $m$ represents the distance from the ray to the world origin. Given a set of 2D pixel

Figure 4: **Pipeline of camera-controlled image-to-video diffusion model.** We follow CameraCtrl to add a learnable pose encoder and a linear projection to process plucker embeddings as a global positional embedding. Epipolar attention is added between spatial and temporal attention.

coordinates $\{(u,v)_i\}^n$, ray directions $d$ can be computed by the unprojection transform:

$$d = R^{-1}K^{-1} \cdot (u,v,1)^{\mathrm{T}}, \ m = (-R^{-1}T) \times d \tag{3}$$

**Text-guided Image to Video Diffusion Model.** Text-guided Image to Video Diffusion Model (Zhang et al., 2024; 2023b; Xing et al., 2023) learn a video data distribution by the gradual denoising of a variable sampled from a Gaussian distribution. For image to video generation, first, a learnable auto-encoder (consisting of an encoder $\mathcal{E}$ and a decoder $\mathcal{D}$) is trained to compress the video into latent space. Then, a latent representation $z = \mathcal{E}(x)$ is trained instead of a video $x$. Specifically, the diffusion model $\epsilon_\theta$ aims to predict the added noise $\epsilon$ at each timestep $t$ based on the text condition $c_{\text{txt}}$ and the reference image condition $c_{\text{img}}$, where $t \in \mathcal{U}(0,1)$. The training objective can be simplified as a reconstruction loss:

$$\mathcal{L} = \mathbb{E}_{z,c_{\text{txt}},c_{\text{img}},\epsilon \sim \mathcal{N}(0,\mathrm{I}),t} \left[ \left\| \epsilon - \epsilon_\theta \left( \mathbf{z}_t, c_{\text{txt}}, c_{\text{img}}, t \right) \right\|_2^2 \right], \tag{4}$$

where $\mathbf{z} \in \mathbb{R}^{F \times H \times W \times C}$ is the latent code of video data with $F, H, W, C$ being frame, height, width, and channel. Besides, $c_{\text{text}}$ is the text prompt for input video, and $c_{\text{img}}$ is the reference frame of video. A noise-corrupted latent code $\mathbf{z}_t$ from the ground-truth $z_0$ is formulated as $\mathbf{z}_t = \alpha_t z_0 + \sigma_t \epsilon$, where $\sigma_t = \sqrt{1 - \alpha_t^2}$, $\alpha_t$ and $\sigma_t$ are hyperparameters to control the diffusion process.

### 3.2 OVERALL PIPELINE

In this section, we present our novel camera-conditioned method for geometry-consistent image-to-video generation, as shown in Fig. 4. We first describe cross-frame epipolar line and discreterized epipolar mask, grounded in the principle of camera projection. Next, we propose epipolar-constrained attention module for the base model in a plug-and-play manner, which effectively make use of feature correlations along epipolar lines. Further, we discuss the situation when epipolar lines of all frames are outside the image plane and introduce register tokens as a simple yet effective fix. Finally, we leverage multiple CFG to balance visual quality and camera pose consistency.

### 3.3 EPIPOLAR ATTENTION FOR NOISED FEATURES TRACKING

**Epipolar line and mask.** The proposed epipolar attention mechanism seeks to establish a connection between frames, as shown on the left-hand side of Fig. 5. Its primary concept involves utilizing the epipolar line as a constraint, which effectively narrows down the potential matching pixels from one target frame to any other frames. For a single pixel at coordinate $(u,v)$ on the $i$-th frame, the corresponding epipolar line $l_{ij} \in \mathbb{R}^3$ on the $j$-th frame can be formulated as:

$$l_{ij}(u,v) = F_{ij} \cdot (u,v,1)^{\mathrm{T}}, \tag{5}$$

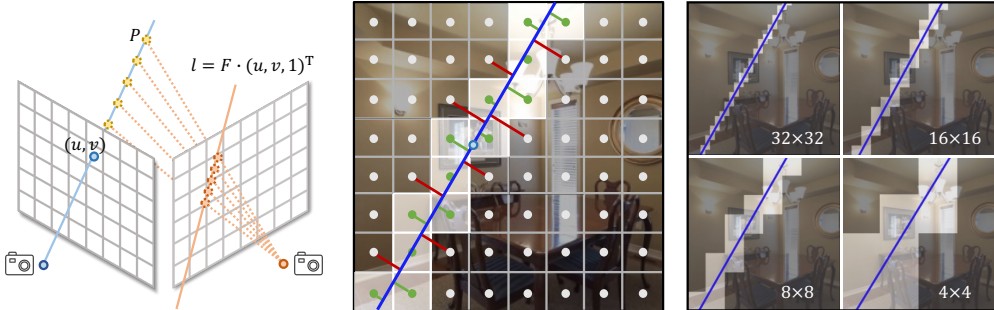

Figure 5: **Epipolar line and mask.** Left: Epipolar constraint of the $j$-th frame from one pixel at $(u, v)$ on the $i$-th frame. Middle: Epipolar mask discretized by the distance threshold $\delta$, so that only neighboring pixels in green are allowed to attend while those red lined are not. Right: Multi-resolution epipolar mask adaptive to the feature size in U-Net layers.

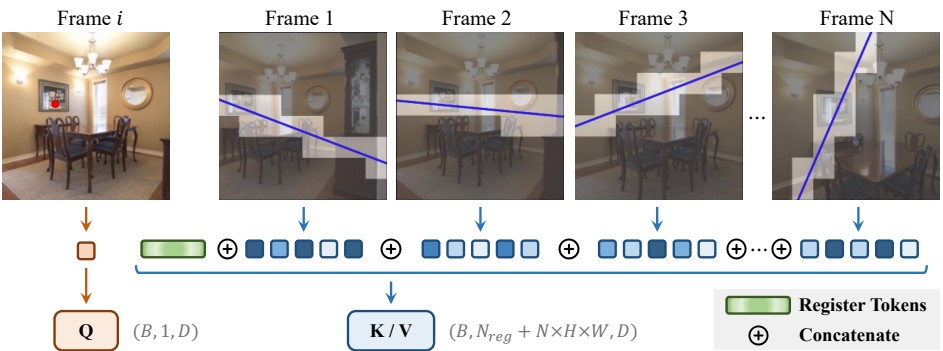

Figure 6: **Epipolar attention mask with register tokens.** We specify query pixel by red point in the $i$-th frame for clarity. Epipolar attention mask is constructed by concatenating epipolar masks along all frames. We insert register tokens to key/value sequence to deal with zero epipolar scenarios.

where $F_{ij}$ is the camera fundamental matrix of two frames, which can be derived as $F_{ij} = K_j^{-\mathrm{T}} \cdot E_{ij} \cdot K_i^{-1}$ given the camera intrinsics $K_i, K_j \in \mathbb{R}^{3\times3}$ and the camera essential matrix $E_{ij} \in \mathbb{R}^{3\times3}$. We transform the camera pose of the $j$-th frame to be relative to the $i$-th frame for simplicity, thus it holds that $E_{ij} = T_{i\to j} \times R_{i\to j}$, where $R_{i\to j} \in \mathbb{R}^{3\times3}$ and $T_{i\to j} \in \mathbb{R}^3$ are the relative rotation matrix and translation vector, respectively. Due to the contiguous representation of the epipolar line $l_{ij} = Ax + By + C$, we convert it to attention mask by calculating per-pixel distance $D$ at coordinate $(u', v')$ on the $j$-th frame to the epipolar line as

$$D_{ij}(u', v') = \frac{(A, B, C) \cdot (u', v', 1)}{\sqrt{A^2 + B^2}}, \tag{6}$$

and filtering out those values that are larger than a threshold $\delta$. We empirically choose half of the diagonal of the feature grid size as the threshold. This approach optimizes the correspondence search space by significantly reducing the number of candidates from $hw$ to $l$, with $l \ll hw$, thereby enhancing efficiency and accuracy.

**Epipolar attention.** We extend current temporal attention with epipolar constraint to leverage cross-frame relationship and inject geometry consistency for video generation.

We denote the query, key and value as $q \in \mathbb{R}^{hw\times c}$, $k \in \mathbb{R}^{Nhw\times c}$ and $v \in \mathbb{R}^{Nhw\times c}$, respectively. Given the epipolar attention mask $m \in \mathbb{R}^{hw\times Nhw}$ introduced in Section 3.3, our epipolar attention that captures relevant contextual information between the $i$-th frame and all $N$ frames is then computed as

$$\mathrm{EpipolarAttn}(q, k, v, m) = \mathrm{softmax}\left(\frac{qk^{\mathrm{T}}}{\sqrt{d}} \odot m\right) v, \tag{7}$$

where $\odot$ denotes Hadamard product and $d$ is the dimension of attention heads for attention score normalization. For detailed computation procedures, please refer to Appendix A.

**Register tokens for scenarios where epipolar lines disappear.** For videos with significant camera movements, dynamic objects, or occlusions, there may be cases where some pixels from the $i$-th frame have no corresponding epipolar lines within the image planes of all $N$ frames. This situation can lead to a zero epipolar mask, affecting the computational stability of the epipolar attention mechanism.

To address this issue, we draw inspiration from Timothée et al. (2024) and introduce additional register tokens to the input sequence as a straightforward solution, as illustrated in Fig. 6. Additionally, register tokens are learnable, enabling adaptive learning to address various special cases. Without register tokens to serve as placeholders, we may encounter the zero length of key/value tokens and fail to calculate attention

### 3.4 MULTIPLE CLASSIFIER-FREE GUIDANCE

**Control for multiple condition.** Similar to DynamicCrafter (Xing et al., 2023; Esser et al., 2023), we introduce two guidance scales $s_{\text{img\&txt}}$ and $s_{\text{camera}}$ to text-conditioned image animation, which can be adjusted to trade off the impact of two control signals:

$$
\begin{aligned}
\hat{\epsilon}_\theta \left( \mathbf{z}_t, \mathbf{c}_{\text{camera}}, \mathbf{c}_{\text{img\&txt}} \right) = {} & \epsilon_\theta \left( \mathbf{z}_t, \mathbf{c}_{\text{camera}}, \varnothing \right) \\
& + s_{\text{img\&txt}} \big( \epsilon_\theta \left( \mathbf{z}_t, \mathbf{c}_{\text{camera}}, \mathbf{c}_{\text{img\&txt}} \right) - \epsilon_\theta \left( \mathbf{z}_t, \mathbf{c}_{\text{camera}}, \varnothing \right) \big) \\
& + s_{\text{camera}} \big( \epsilon_\theta \left( \mathbf{z}_t, \mathbf{c}_{\text{camera}}, \mathbf{c}_{\text{img\&txt}} \right) - \epsilon_\theta \left( \mathbf{z}_t, \varnothing, \mathbf{c}_{\text{img\&txt}} \right) \big).
\end{aligned}
\tag{8}
$$

**Multiple scale distillation for acceleration.** If needed, we can distill (Xing et al., 2023) the two guidance scales $s_{\text{img\&txt}}$ and $s_{\text{camera}}$ into the model to further avoid the extra inference time brought by three times of forward:

$$
\epsilon_\theta \left( \mathbf{z}_t, \mathbf{c}_{\text{camera}}, \mathbf{c}_{\text{img\&txt}}, s_{\text{camera}}, s_{\text{img\&txt}} \right) = \hat{\epsilon}_\theta \left( \mathbf{z}_t, \mathbf{c}_{\text{camera}}, \mathbf{c}_{\text{img\&txt}} \right)
\tag{9}
$$

## 4 METRICS AND EVALUATION

In this section, we present our reproducible evaluation pipeline. Previous studies have employed various evaluation protocols, resulting in inconsistent metrics due to the lack of a common benchmark. The structure-from-motion (SfM) method such as COLMAP (Schonberger & Frahm, 2016), struggles to produce stable and accurate predictions when applied to generated videos. This challenge arises because SfM relies on SIFT operators for keypoint identification, which can lead to erroneous matches when assessing generated content. Such inaccuracies may result in unsolvable equations or significantly flawed estimates of camera extrinsics. Contributing factors include the low resolution of these videos (256x256), the presence of dynamic scenes, the absence of true 3D consistency, and issues related to lighting variations and object distortion.

To address these limitations, we adapt the global structure-from-motion method GLOMAP (Pan et al., 2024) to validate camera pose consistency. Our evaluation pipeline comprises three steps: feature extraction, exhaustive matching, and global mapping. To enhance robustness, we share GT priors for camera intrinsics ($f_x$, $f_y$, $c_x$, $c_y$) and allow the structure-from-motion process to focus primarily on optimizing camera extrinsics. Detailed CLI parameters can be found in Appendix B.

Before calculating metrics, we canonicalize the estimated camera-to-world matrices by converting each frame relative to the first frame and normalizing the scene scale using the $\mathcal{L}_2$ distance from the first camera to the furthest cameras. To account for randomness introduced by GLOMAP, we conduct five individual trials for each of the 1,000 sampled videos, averaging only those trials that are successful per sample. The final metrics, including RotError, TransError, and CamMC, are averaged on a sample-wise basis.

**RotError (He et al., 2024a).** We evaluate per-frame camera-to-world rotation accuracy by the relative angles between ground truth rotations $R_i$ and estimated rotations $\tilde{R}_i$ of generated frames. We report accumulated rotation error along 16 frames in radians.

$$
\text{RotErr} = \sum_{i=1}^{n} \cos^{-1} \frac{\text{tr}(\tilde{R}_i R_i^{\mathrm{T}}) - 1}{2}
\tag{10}
$$

Table 1: **Quantitative comparison with state-of-the-art methods.** * denotes the results we reproduced using DynamiCrafter as base I2V model. We achieve a 32.96%, 25.64%, 20.77% improvement over previous Sota CameraCtrl on RotErr, CamMC, TransErr on the RealEstate10K dataset without compromising dynamics, generation quality, and generalization on out-of-domain images. These results were obtained using Text and Image CFG set to 7.5, 25 steps, and camera CFG set to 1.0 (no camera cfg).

| Method | Publication | TransErr ↓ | RotErr ↓ | CamMC ↓ | FVD ↓ | |
| | | | | | VideoGPT | StyleGAN |
|---|---|---|---|---|---|---|
| DynamiCrafter (Xing et al., 2023) | ECCV 2024 | 9.8024 | 3.3415 | 11.625 | 106.02 | 92.196 |
| + MotionCtrl (Wang et al., 2024d)* | SIGGRAPH 2024 | 2.5068 | 0.8636 | 2.9536 | 70.820 | 60.363 |
| + CameraCtrl (He et al., 2024a)* | arXiv 2024 | 1.9379 | 0.7064 | 2.3070 | 66.713 | 57.644 |
| + CamI2V (Ours) | | **1.4955** | **0.4758** | **1.7153** | **66.090** | **55.701** |

**TransError (He et al., 2024a)**. We evaluate per-frame camera trajectory accuracy by the camera location in the world coordinate system, i.e. the translation component of camera-to-world matrices. We report the sum of $\mathcal{L}_2$ distance between ground truth translations $T_i$ and generated translations $\tilde{T}_i$ for all 16 frames.

$$\text{TransErr} = \sum_{i=1}^{n} \left\| \tilde{T}_i - T_i \right\|_2 \tag{11}$$

**CamMC (Wang et al., 2024d)**. We also evaluate camera pose accuracy by directly calculating $\mathcal{L}_2$ similarity of per-frame rotations and translations as a whole. We sum up the results of 16 frames.

$$\text{CamMC} = \sum_{i=1}^{n} \left\| \left[ \tilde{R}_i | \tilde{T}_i \right] - \left[ R_i | T_i \right] \right\|_2 \tag{12}$$

**FVD (Unterthiner et al., 2018)**. Additionally, to ensure that proposed method coherently improve generative capability and visual quality of base I2V model, we evaluate the distance of generated frames from training distribution by Fréchet Video Distance (FVD).

## 5 EXPERIMENTS

### 5.1 SETUP

**Dataset.** We train our model on RealEstate10K (Zhou et al., 2018) dataset, which contains approximately 70K video clips at the resolution of around 720P with camera poses annotated by SLAM-based methods. We resize video clips from dataset to 256 while keeping the original aspect ratio and perform center cropping to fit in our training scheme. We sample 16 frames from single video clip when training with a random frame stride ranging from 1 to 10. We set fixed frame stride of 8 for inference. We take random condition frame for generation as data augmentation.

**Implementation Details.** We choose DynamiCrafter (Xing et al., 2023) as our base image-to-video (I2V) model and implement proposed method on the top of it. For fair comparision, we also make reproduction work of MotionCtrl (Wang et al., 2024d) and CameraCtrl (He et al., 2024a), since their public accessible versions are either T2V or SVD-based. We project Plücker embedding into base model by a pose encoder similar to the architecture in CameraCtrl. We freeze all parameters from base model and train proposed method at the resolution of 256×256. We set 2 register tokens for the epipolar module to attend when no relevant pixels are on the epipolar line. We apply the Adam optimizer with a constant learning rate of $1 \times 10^{-4}$. We follow DynamiCrafter to choose Lightning as our training framework with mixed-precision fp16 and DeepSpeed ZeRO-1. We train proposed method and variants on 8 NVIDIA 3090 GPUs with effective batch size of 64 for 50K steps.

### 5.2 QUANTITATIVE COMPARISON

We compare our CamI2V with the latest methods in camera controlled image-to-video generation, including DynamiCrafter (Xing et al., 2023), MotionCtrl (Wang et al., 2024d) and CameraCtrl (He et al., 2024a). As reported in Table 1, our CamI2V significantly improves the camera controllability and visual quality, with substantial reductions in RotErr, TransErr, CamMC and FVD. Compared

Table 2: **Ablation study on model variants.** ◯ denotes our implementation of epipolar attention only on reference frame, similar to CamCo. Our proposed method (Plücker embedding along with epipolar attention on all frames) achieves SOTA performance among all variants.

| Method | Plücker | Epipolar | 3D Full | TransErr↓ | RotErr↓ | CamMC↓ | FVD↓ VideoGPT | StyleGAN |
|---|---|---|---|---|---|---|---|---|
| DynamiCrafter +CamI2V (Ours) | ✓ | ✓ | | **1.4955** | **0.4758** | **1.7153** | 66.090 | **55.701** |
| | ✓ | ◯ | | 1.6014 | 0.5738 | 1.8851 | 66.439 | 56.778 |
| | ✓ | | ✓ | 1.8215 | 0.6299 | 2.1315 | 71.026 | 60.000 |
| | ✓ | | | 1.8877 | 0.7098 | 2.2557 | **66.077** | 55.889 |
| | | ✓ | | 5.5119 | 1.3988 | 6.2855 | 92.605 | 81.447 |
| DynamiCrafter | | | | 9.8024 | 3.3415 | 11.625 | 106.02 | 92.196 |

to CameraCtrl, our method reduces RotErr by 0.2306, translating to a 13.21° decrease in rotational error, which marks a significant improvement. And our method surpasses the state-of-the-art method CameraCtrl in other camera controllability and FVD metrics.

## 5.3 ABLATION STUDY

Adding more conditions to generative models typically reduces uncertainty and improves generation quality (e.g. providing detailed text conditions through recaption). In this paper, we argue that it is also crucial to consider *noisy conditions* like latent features $z_t$, which contain valuable information along with random noise. For instance, in SDEdit (Meng et al., 2021) for image-to-image translation, random noise is added to the input $z_0$ to produce a noisy $z_t$. The clean component $z_0$ preserves overall similarity, while the introduced noise leads to uncertainty, enabling diverse and varied generations.

In this paper, we argue that **providing the model with more noisy conditions, especially at high noise levels, does not necessarily reduce more uncertainty, as the noise also introduces randomness and misleadingness**. This is the key insight we aim to convey.

To validate this point, we designed experiment with the following setups:

1. **Plücker Embedding (Baseline)**: This setup, akin to CameraCtrl, has minimal noisy conditions on cross frames due to the inefficiency of the indirect cross-frame interaction (spatial and temporal attention).

2. **Plücker Embedding + Epipolar Attention only on reference frame**: Similar to CamCo, this setup treats the reference frame as the source view, enabling the target frame to refer to it. It accesses **a small amount** of noisy conditions on the reference frame. However, some pixels of the current frame may have no epipolar line interacted with reference frame, causing it to degenerate to a CameraCtrl-like model without epipolar attention.

3. **Plücker Embedding + Epipolar Attention (Our CamI2V)**: This setup can impose epipolar constraints with all frames, including adjacent frames that have interactions in most cases to ensure an sufficient amount of noisy conditions.

4. **Plücker Embedding + 3D Full Attention**: This configuration allows the model to directly interact with features of all other frames, accessing the most noisy conditions.

**The amount of accessible noisy conditions of the above four setups increase progressively**. One might expect that 3D full attention, which accesses the most noisy conditions, would achieve the best performance. However, as shown in Tab. 2, 3D full attention performs only slightly better than CameraCtrl and is inferior to CamCo-like setup who only applies epipolar attention on reference frame. Notably, our method achieves best result by interacting with more noisy conditions along the epipolar lines. It can be clearly seen in the comparison part in supplementary that CamCo-like setup reference much on the first frame and cannot generate new objects. The 3D full attention generates objects within large movement due to its access to all frames pixels while it is affected by incorrect position of pixels. These findings confirm our insight that **an optimal amount of noisy conditions leads to better uncertainty reduction, rather than merely increasing the quantity of noisy conditions.**

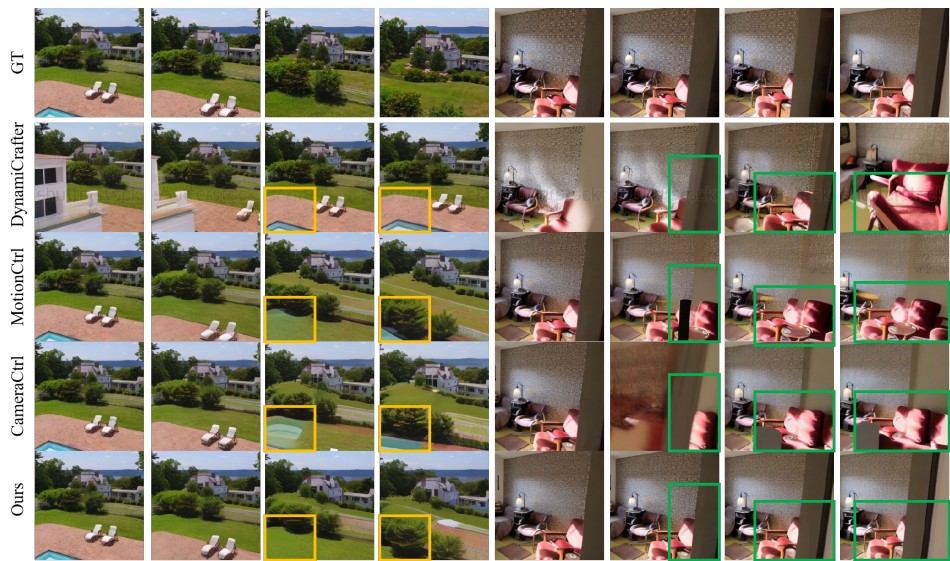

Figure 7: Qualitative Comparison on RealEstate10K.

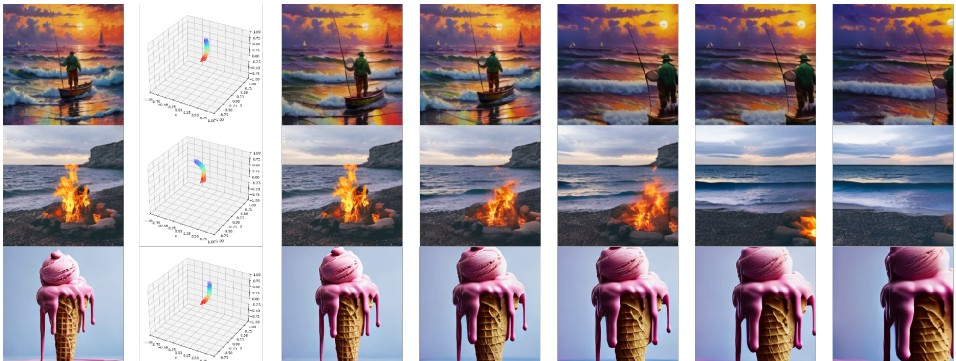

Figure 8: Out-of-Domain Visualization.

## 5.4 QUALITATIVE COMPARISON

**Visualization on RealEstate10K.** As shown in Fig. 7, we present the visualization results of DynamiCrafter, MotionCtrl, CameraCtrl and our CamI2V. It can be observed that the camera trajectory of our method aligns more closely with GT compared with other methods, and the rendering of certain details appears more realistic in the video generated by our CamI2V.

**Out-of-domain visualization.** Our CamI2V demonstrates strong generalization capabilities, enabling direct application to camera controlled video generation across out-of-domain content, such as oil paintings, photography, and animation, as shown in Fig. 8.

## 6 CONCLUSION

In this paper, we address the integration of camera poses into diffusion models to enhance their understanding of the physical world in text-guided image-to-video generation. We propose a novel framework utilizing Plücker coordinates as 3D ray embeddings and introduce an epipolar attention mechanism that aggregates features along epipolar lines, ensuring robust tracking even under high noise conditions. Additionally, we incorporate register tokens to manage scenarios where frames lack intersections due to rapid camera movements or occlusions. Our methods significantly improve controllability and stability, achieving state-of-the-art performance on RealEstate10K and out-of-domain datasets. However, challenges remain in high-resolution generation, handling complex camera trajectories, and maintaining generation quality in long videos. Future work will focus on these aspects, alongside releasing checkpoints and training/evaluation codes to support further research.

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

## A   CORE CODES

---

**Algorithm 1** Spatial Attention Block

---

**Require:** U-Net feature $x$, condition $c$
1: $x \leftarrow x + \text{SelfAttn}_1(\text{PreNorm}(x))$
2: $x \leftarrow x + \text{CrossAttn}_2(\text{PreNorm}(x), c)$
3: $x \leftarrow x + \text{FFN}(\text{PreNorm}(x))$
4: **return** $x$

---

**Algorithm 2** Temporal Attention Block with Camera Control

---

**Require:** U-Net feature $x$, condition $c$, plücker embedding $p$, epipolar attention mask $m$
1: $x \leftarrow x + \text{Linear}(\text{PreNorm}(x) + \text{PreNorm}(p))$ $\qquad\qquad$ ▷ Pücker Ray Embeddings
2: $x \leftarrow x + \text{EpipolarAttn}(\text{PreNorm}(x), m)$
3: $x \leftarrow x + \text{SelfAttn}_1(\text{PreNorm}(x))$
4: $x \leftarrow x + \text{SelfAttn}_2(\text{PreNorm}(x))$
5: $x \leftarrow x + \text{FFN}(\text{PreNorm}(x))$
6: **return** $x$

---

**Algorithm 3** Epipolar Attention Mask

---

**Require:** Intrinsic matrices $K$, extrinsic matrices $[R|T]$, feature size $H \times W$, threshold $\delta$
1: $E \leftarrow T \times R$ $\qquad\qquad\qquad\qquad\qquad\qquad\qquad\qquad\qquad$ ▷ Essential matrices $E$
2: $F \leftarrow K^{-\text{T}} \cdot E \cdot K^{-1}$ $\qquad\qquad\qquad\qquad\qquad\qquad$ ▷ Fundamental matrices $F$
3: $g \leftarrow \text{mesh\_grid}(H, W)$ $\qquad\qquad\qquad\qquad$ ▷ Homogeneous feature coordinates $g$
4: $l \leftarrow \text{normalize}(F \cdot g^{\text{T}})$ $\qquad$ ▷ Epipolar line $l = Ax + By + C$, normalized by $\sqrt{A^2 + B^2}$
5: $d \leftarrow l^{\text{T}} \cdot g^{\text{T}}$ $\qquad\qquad\qquad\qquad$ ▷ Distance $d$ from feature coordinates to epipolar lines
6: $m \leftarrow [\text{reg}] \oplus \text{flatten}(d < \delta)$ $\qquad\qquad\qquad\qquad$ ▷ Epipolar attention mask $m$
7: **return** $m$

---

## B   COLMAP & GLOMAP CONFIGURATION

We assume `SIMPLE_PINHOLE` as the common camera model for all video clips and all 16 frames from the same video clip share the same camera intrinsics. For the feature extractor, we enable `estimate_affine_shape` and `domain_size_pooling` in `SiftExtraction`, while fix camera intrinsics by passing ($f_x$, $f_y$, $c_x$, $c_y$) into `ImageReader.camera_params`. For the exhaustive matcher, we enable `guided_matching` and set `max_num_matches` to 65536 in `SiftMatching` to make possible more underlying matches. For the global mapper, we disable `BundleAdjustment.optimize_intrinsics` and relax the geometric constraint by extending `RelPoseEstimation.max_epipolar_error` to 4.

## C   GPU MEMORY AND SPEED

Table 3: **Comparison on GPU memory usage and speed under DeepSpeed ZeRO-1.** * denotes our reproduction on DynamiCrafter. We report full parameter fine-tuning results of DynamiCrafter. Our model can be trained on 24GB consumer-level GPUs despite the additional epipolar attention.

| Method | # Params Trainable | GPU Memory (GiB) ↓ | | Time (s) ↓ | | |
| --- | --- | --- | --- | --- | --- | --- |
| | | Inference | Training | Forward | Backward | Optimizer |
| DynamiCrafter | 1.4 B | 11.14 | 23.72 | 0.413 | 0.856 | 1.959 |
| DynamiCrafter+MotionCtrl* | 63.4 M | 11.18 | 16.75 | 0.387 | 0.198 | 0.636 |
| DynamiCrafter+CameraCtrl* | 211 M | 11.56 | 18.44 | 0.398 | 0.247 | 0.723 |
| DynamiCrafter+CamI2V (Ours) | 261 M | 11.67 | 21.71 | 0.403 | 0.458 | 0.974 |

# D    EXTRA OUT-OF-DOMAIN VISUALIZATIONS

Dynamic videos are best viewed at our **local anonymous web page**. It's strongly recommended to view the visualizations in the supplementary for a more comprehensive evaluation.

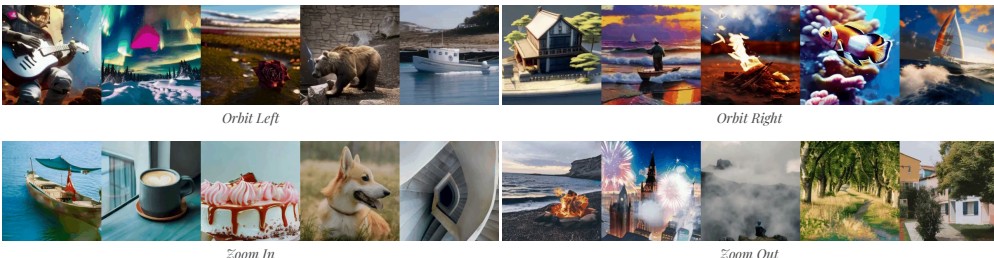

Figure 9: Visualization of our 256×256 model.

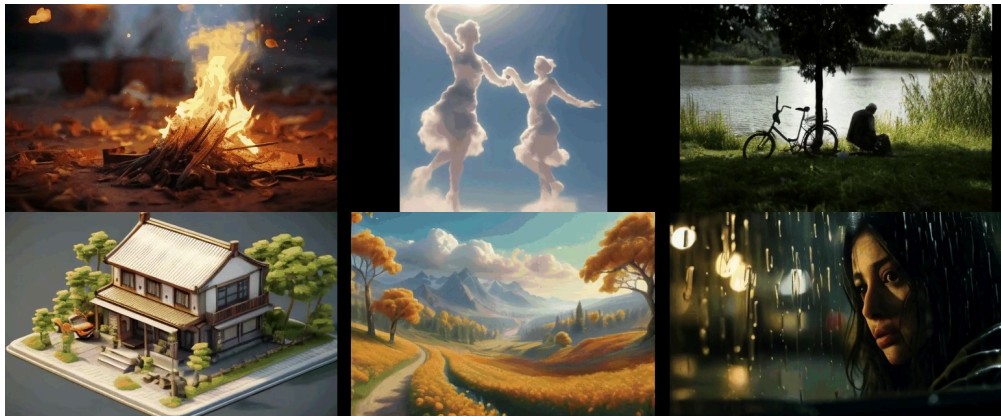

Figure 10: Visualization of original outputs from our 512×320 model, with no padding removed.

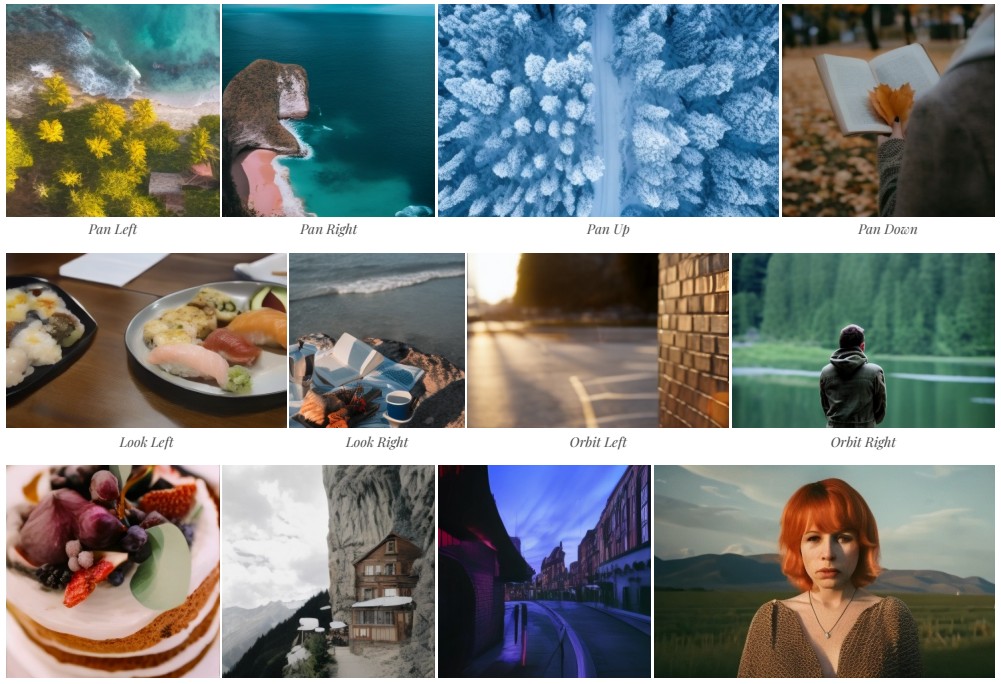

Figure 11: Generated by our 512×320 model, compatible with input images of arbitary aspect ratio.

