# OpenReview forum: "CamI2V: Camera-Controlled Image-to-Video Diffusion Model"
_ICLR.cc/2025/Conference — Submitted to ICLR 2025_

### Official Review · Reviewer_iyw3 · 2024-11-01

**Soundness:** 3
**Presentation:** 3
**Contribution:** 2
**Rating:** 5
**Confidence:** 4

**Summary:**

This paper retrofits the text-guided video diffusion model with a camera control module, enhancing the camera trajectory controllability and 3D consistency for image animation.

By representing a pinhole camera model as a bundle of ray-level 3D embeddings through Pl&uuml;cker coordinates, the denoising process is conditioned on camera poses in a _clean_ and expressive manner. To facilitate multi-view 3D reasoning and efficiency, the epipolar attention mechanism is developed, which only aggregates cross-view information within a limited region along the epipolar line.

The authors test the method on RealEstate10K and out-of-domain datasets using an improved evaluation pipeline, and demonstrate its superiority over other alternatives.

**Strengths:**

- The idea of introducing epipolar geometry to the temporal attention is straightforward and well-motivated, which allows the model to explicitly reason about the 3D world.
- It is interesting and technically sound to configure register tokens for degenerate cases where no corresponding epipolar lines are detected.
- Experiments on RealEstate10K and in the wild have shown that the proposed method improves the camera motion controllability and stability.

**Weaknesses:**

- One major concern is that the proposed method is somewhat lack of novelty. The epipolar attention mechanism is not new [1], while the Pl&uuml;cker embedding is adopted from [2].  There are some impressive innovations, e.g. the concept of _clean/noised conditions_ and the register tokens, but related exposition and analysis could be more thorough. Please see Q1-5 in the question section for details.
- In the conclusion, the authors announce that their method _significantly_ improves the controllability and stability (Line 534-535), which may be an overstatement. In Tab.1, the advantage of CamI2V (the proposed method) over CameraCtrl is kind of minor, and in Tab.2, the introduction of the epipolar attention quantitatively appears to bring in only limited improvement. It would be nice to include additional (qualitative) results that demonstrate the effectiveness of each contribution. Also see Q6.
- One contribution that the authors adapt GLOMAP for more robust evaluation pipeline raises another concern. It is not convincing that GLOMAP can effectively address limitations such as low resolution and dynamic scenarios. More details and experimental comparisons would be helpful to demonstrate the merits of the proposed evaluation protocol.

[1] Hung-Yu Tseng, et al. "Consistent View Synthesis with Pose-Guided Diffusion Models." Proceedings of the IEEE/CVF Conference on Computer Vision and Pattern Recognition (CVPR). 2023.

[2] Hao He, et al. "CameraCtrl: Enabling Camera Control for Text-to-Video Generation." arXiv preprint arXiv:2404.02101. 2024.

**Questions:**

1. Elaboration on the insight of the _clean/noised conditions_. The Pl&uuml;cker embeddings and the proposed epiplolar attention are regarded as clean conditions, but no further clarification is given. It would be preferable to include some mathematical analysis on how clean conditions can reduce uncertainty more effectively than noisy conditions, as the current perspective seems just intuitive and empirical rather than theoretically supported.
2. I appreciate the idea of reserving register tokens for special cases, but the authors fail to demonstrate its effectiveness through essential ablation studies.
3. In the epipolar attention part, a hard mask strategy is employed, which filters out pixels far from epipolar lines, while [1] proposes the epipolar weight matrix (a form of soft masks). It would be interesting to draw a comparison between them. The hard mask strategy might be computationally more efficient, so for example, more specific details about the computational complexity or running time would be beneficial.
4. The epipolar geometry is based on the assumption that observations are all static. Wouldn't the proposed epipolar attention incur degradation in cases involving dynamic objects?
5. The multiple guidance scale is not only similar to [2] but also lacks explanation and experimental support. The authors should reconsider whether to count it as one contribution (Line 133-134).
6. Limited qualitative results. Camera trajectories used in supplementary materials are similar and relatively simple. Diverse camera trajectories for evaluating are in great need.

- The authors may have to check the citation in Line 208-209: we follow _(Tseng et al., 2023)_ to represent cameras as ray bundles.
- In the top middle of Fig. 4, what do the "Project" module and its learnable weights refer to?
- Please check the format of notations. Make sure they are consistent throughout the paper. For example, the latent code should be in bold: $\mathbf{z}_t$.

[1] Hung-Yu Tseng, et al. "Consistent View Synthesis with Pose-Guided Diffusion Models." Proceedings of the IEEE/CVF Conference on Computer Vision and Pattern Recognition (CVPR). 2023.

[2] Jinbo Xing, et al. "DynamiCrafter: Animating Open-domain Images with Video Diffusion Priors." Proceedings of the 18th European Conference on Computer Vision (ECCV). 2024

---

> ### Author Response · Authors · 2024-11-24
> **Response to controllability improvement, evaluation benchmark, more visualizations results, dynamic issuses, , soft mask.**
>
> # Response to  Performance Improvement
> Please refer to the **Table** we provided in general response in **To all reviwers: 1. The Connection Between Our Insights on Conditions and the Proposed Method” for additional context.**
>
> Our method demonstrates significant improvements over CameraCtrl, achieving a **32.96% reduction in Rotation Error**, a **25.64% decrease in CamMC**, and a **20.77% improvement in Translation Error**, **without any decrease in FVD**. These results were obtained using Text and Image CFG set to 7.5, 25 steps, and **camera CFG set to 1.0 (no camera cfg)**.
>
> # Response to Evaluation Benchmark Improvement
> We have identified that current evaluation benchmarks are unreliable, and no open-source evaluation code is available. To address this, we propose several improvements:
> * Performing multiple measurements (averaging over five runs) to ensure consistency.
> * Conducting extensive evaluations on a large number of samples (1000 samples) for statistical significance.
> * Replacing the SfM algorithm with a more robust GLOMAP (**new work of COLMAP's author, small improvement**) to improve accuracy.
> * Computing relative poses with respect to the initial frame to standardize evaluations.
> * Introducing normalization to address the issue of unknown global scene scale caused by SfM predictions (**the most important improvement**).
>
> We also commit to open-sourcing all our training and evaluation code, as well as the checkpoints. Although these modifications are small and straightforward, they hold significant importance for improving reliability and reproducibility in the community.
>
>
> # Response to Q6: More Qualitative Results.
> Please refer to **provided supplementary, and open the index.html.**
>
> # Response to Q4: Potential Dynamics Drecrease
> Please refer to **provided supplementary, and open the index.html.** Dynamics in generated videos are quite good.
>
> Only applying epipolar attention without original spatial+temporal attention will lead to dynmaics decrease.
> Therefore, we freeze pretrained model and add extra epipolar attention,
> functioning as a supplement for direct cross-frame interaction in an effective way
> with access to an appropriate amount of noisy conditions along the epipolar lines.
>
> # Response to Q5: Multiple CFG
> Thank you for your valuable suggestion. We have already cited Dynamicrafter in our submission.
> The multiple CFGs include camera control and represent a minor inference technique.
> Following your advice, we will revise it as a minor contribution.
>
> # Response to Q3: Soft Mask vs. Hard Mask
> We are grateful for the insightful suggestion to apply soft masks and find it to be an excellent idea.
> Although we have read relevant literature on soft masking, we opted not to incorporate it before to maintain simplicity and avoid introducing additional equations.
>
> **The running time remains consistent** since all methods utilize dynamic attention masks in current optimized memory-efficient attention frameworks. They are only slightly slower than attention mechanisms without masks.
>
> | Method                  | RotErr$\downarrow$ | TransErr$\downarrow$ | CamMC$\downarrow$ | FVD (VideoGPT)$\downarrow$ | FVD (StyleGAN)$\downarrow$ |
> |-----------------|---------------|---------------|---------|---------|---------|
> | DynamiCrafter            | 3.3415             | 9.8024       | 11.625   | 106.02      | 92.196     |
> | +MotionCtrl          | 0.8636             | 2.5068         | 2.9536            | 70.820               | 60.363     |
> | +Plucker Embedding (Baseline, CameraCtrl)    | 0.7098             | 1.8877               | 2.2557     | 66.077       | 55.889      |
> | +Plucker Embedding + Epipolar Attention Only on Reference Frame (CamCo-like) | 0.5738    | 1.6014               | 1.8851            | 66.439     | 56.778      |
> | +Plucker Embedding + Epipolar Attention (Our CamI2V)    | **0.4758**         | 1.4955      | 1.7153        | 66.090       | **55.701**     |
> | +Plucker Embedding + Epipolar Attention (**Soft Mask**)   | 0.5051             | **1.4667**   | **1.7147**     | **65.519**  | 56.159     |
> | +Plucker Embedding + 3D Full Attention   | 0.6299             | 1.8215   | 2.1315     | 71.026  | 60.00     |
>
> We define attention bias as $log(exp(-d * temperature ))$ and add attention bias into attention logits, where $d$ represents distance to the epipolar line or plane. The application of soft masks results in only a slight improvement, which may be attributed to the limited time available for adjusting the temperature hyperparameters. We anticipate that with further optimization, soft masks could yield better results.

---

> ### Author Response · Authors · 2024-11-24
> **Response to Elaboration on the insight of the clean/noised conditions.**
>
> # Response to Q1: Elaboration on the insight of the clean/noised conditions.
> Please also refer to **our general response titled “1. The Connection Between Our Insights on Conditions and the Proposed Method” for additional context.**
>
> Clean conditions do not necessarily reduce uncertainty more effectively than noisy conditions. Their effectiveness depends on the information provided and is often related to the amount of condition information. For example, recaptioning in Stable Diffusion 3 generates more detailed text captions (clean conditions), resulting in better generation quality. In image-to-video generation settings, noisy conditions (noisy latent features such as $z_t$ typically originate from user-provided reference images. If  $z_0$ is not provided by users, noisy conditions also arise from $z_t$ generated in previous timesteps. The noisy features $z_t$ have a shape of  $F \times C \times H \times W$, which is significantly larger than clean conditions (e.g., text, which contains only textual information). When the added noise does not dominate the valid information, noisy conditions can reduce uncertainty more effectively by providing extensive details, colors, layout, and other relevant information.
>
> Previous methods have introduced epipolar attention primarily with the motivation of enhancing geometric consistency, **but these approaches were not specifically designed for settings with inherent noise**. In contrast, our work focuses on the unique challenges posed by the noisy feature spaces inherent in diffusion models.
>
> Our motivation is to understand **how noise affects cross-frame interactions** and to design strategies that mitigate the misleading effects of noise. In clean feature spaces, issues like feature copying due to large receptive fields can be mitigated by increasing model capacity and improving feature extraction. The main challenge in such settings is incorrect attention allocation when the aggregation space is too large.
>
> However, **in noisy feature spaces, the situation is different**. Due to the added noise, dissimilar features may erroneously receive high attention scores. In this scenario, even powerful models struggle to reconstruct clean pixels from random noise—especially at high noise levels—because the noise can mislead the attention mechanism, resulting in degraded performance.
>
> Our novel insight is that **simply increasing the receptive field, such as using 3D full attention, leads to a greater amount of noisy conditions.** Due to the randomness inherent in the noise, **accessing more noisy conditions does not necessarily reduce uncertainty and thus not necessarily lead to better performance**. This understanding suggests that models should focus on aggregating information from more reliable locations to effectively reduce uncertainty caused by noise. The idea of **uncertainty**  introduced by noisy features but not clean features leads to a new insight.
>
> **We have found that applying epipolar constraints is one of the most suitable way to prevent the model from being misled by noise.** By restricting attention to features along the epipolar lines, the model can interact with more relevant and **less noisy information**, improving cross-frame interactions in diffusion models. We believe these new insights are significant as they may inspire researchers to develop new architectural designs that better handle noisy conditions in generative models.

---

> ### Author Response · Authors · 2024-11-24
> **Response to Design for Epipolar Attention when Epipolar Lines Vanish**
>
> ## Response to Q2: Design for Epipolar Attention when Epipolar Lines Vanish
>
>
> Please also refer to our general response titled **4. Regarding the Register Token** for additional context.
>
> We apologize for not clearly presenting the details regarding this issue earlier.
> When we attempted to **apply epipolar attention without register tokens, we encountered NaN** errors during training and **were unable to successfully train a model for ablation studies**.
>
> Our novelty lies in **consideration of  situations without epipolar lines**,
> which is crucial for robust performance in practical applications. Regarding this scenario, we have observed that **previous works on epipolar attention often do not provide special designs or detailed descriptions for handling cases where epipolar lines are absent.** This gap motivated us to develop an effective solution.
>
> **When epipolar line exists but is far away from the queried frame, the zero length kv tokens in memory-efficient attention leads to NaN during training, making it impossible to train a model.** In this case, **two added register tokens serve as placeholder** to ensure the non-zero length of kv tokens.
>
> Despite utilizing only two register tokens is insufficient, it **ensures the model's basic functionality and only part of pixels on the current frame who have no epipolar line degrade to CameraCtrl without epipolar attention.**

---

> ### Comment · Reviewer_iyw3 · 2024-11-25
> **Feedback to Authors' Response**
>
> I have thoroughly read the authors' response as well as the feedback from other reviewers. I appreciate the authors' efforts in addressing our concerns.
>
> Some of the key messages that I find particularly helpful to my major concerns:
> - Elaboration on the insight of the clean/noised conditions. To my understanding, by introducing the concept of clean/noised conditions, the authors aim to demonstrate that the widely adopted epipolar attention mechanism is likely one of the most suitable way to handle noised cross-frame interactions. More specifically, applying epipolar constraints across frames could include adequate noised features (large noises) while preventing deterministic information from being obscured by noise. This interpretation is supported by the provided quantitative results and currently makes sense to me.
> - Regarding the register token. The authors state in their response that the register token is essential for training stability. Without register tokens, applying epipolar attention becomes infeasible, and thus no ablation.
> - Regarding evaluation metrics. From my perspective, the contribution here is not the use of GLOMAP but the commitment to open-sourcing the evaluation code.
>
> Some concerns remain unresolved, and further clarification is needed:
> - I notice that the quantitative results in the updated revision differ from those in the original submission. The authors should explicitly clarify this distinction in their response. In Tab. 1, why is there a performance drop in the "DynamiCrafter+CameraCtrl" method? The response to performance improvement seems questionable.
> - In the top middle of Fig. 4, what do the "Project" module and its learnable weights refer to?

---

> ### Author Response · Authors · 2024-11-25
> **Response to different quantitative results and "Project" Module.**
>
> We sincerely appreciate your valuable feedback.
>
> ## Response to different quantitative results
>
> To quickly finish the experiments, we change to six machines equipped with eight H100 GPUs and reproduce all the experiments.
> We also train from scratch 512x320 model for generating visualizations in index.html.
>
> The primary factor contributing to performance differences was that, in the initial draft, our CameraCtrl and MotionCtrl models employed a fixed condition frame index, consistently using the first frame as the reference image. In contrast, our CamI2V model is designed to support any frame as the reference by using a random condition frame index during training.
>
> To ensure a fair and transparent comparison between the methods, the latest version of our submission uniformly employs a random condition frame index across all models. Additionally, to mitigate the influence of camera CFG, we set the camera CFG to 1.0 (no camera CFG) while in our first submission we use camera cfg 1.875, where an overemphasis on camera controllability may result in a substantial decrease in generation quality.
>
> We believe these changes provide a more equitable evaluation of our proposed method and clearly highlight the differences between the approaches.
>
>
> ## Response to "Project" Module
> We follow the "Projection" module of CameraCtrl, which functions as a linear projection to align the channels of the Plücker embedding output from pose encoder before adding to unet features. In CameraCtrl paper, it is described that "A learnable linear layer is adopted to further fuse two representations which are then fed into the first temporal attention layer of each temporal block" and it can be seen in the pipelilne image(fig 2 in their paper).

---

> ### Comment · Reviewer_iyw3 · 2024-12-02
>
> I thank the authors for their response and have read through the revised draft.
>
> Regarding W1, the authors reconfirm that their primary contribution lies in the effective modeling of noisy cross-frame interactions. While it is well-motivated and somewhat makes sense, I have to maintain my concern as the current interpretation still seems intuitive and not sufficiently convincing to me. As the primary contribution, more technical insights or findings supporting the proposed concept, such as intermediate observations and informative comparisons, are definitely essential. This is not something that can solved with minor edits and clarification alone but needs restructuring and further investigations.
>
> I'll keep my rating.

---

### Official Review · Reviewer_yhFg · 2024-11-03

**Soundness:** 3
**Presentation:** 3
**Contribution:** 1
**Rating:** 5
**Confidence:** 5

**Summary:**

This paper introduces CAMI2V, a camera controlled I2V model that integrates several camera-based components, such as epipolar attention and plucker embeddings, image conditions and text prompts.

**Strengths:**

The paper is easy to follow, figures are intuitive and look good.

Experiments shows the model outperforms all available state-of-the-art works.

**Weaknesses:**

Novelty of the paper: The contribution of this paper highly resembles CamCo, which is released in June (3.5 months prior to the submission deadline). Both methods use plucker embedding, epipolar lines, and are aimed for image-2-video model. The paper mentioned that CamCo does not supports video trajectories with non-overlapping frames, and the introduction of register token alleviate this problem. Yet this can be  considered as a relatively minor improvement, and is not well supported by experiments. While CamCo does not release its code for reproduction, an ablation study on the token (comparing to other trivial techniques, such as averaging all K/V together), might be helpful to show its effectiveness. Aside from CamCo, CVD (Kuang et.al. 2024) also applies the epipolar attention in its cross-view modules, and CameraCtrl/CVD both use plucker embedding. All these methods challenge the novelty of this work.

While many of the components proposed by the paper are originated from other prior works, the paper spent a huge portion of space to explain these components, overcomplicating the model itself. For example, in Figure 1/2 the authors try to analyze the differences between image/text conditions and noisy latents and show the importance of the epipolar attention, and later provides very detailed calculation on how to compute the attention maps. These contents are somehow redundant since they are already been proposed in prior works.

The author also claims the improvement of the evaluation protocol as one of the major contributions, yet the changes are rather trivial. It only replaces COLMAP with GLOMAP, and fixing the GT camera parameters to the registration.

All of the examples shown in the paper are in relatively small camera changes, which does not support the claims that CAMI2V can handle large camera movements with non-overlapped area. One of the example in Ours_I2V_on_unseen_image_3.gif also shows strong inconsistency across frames (5’th one from the left)

**Questions:**

See weakness

---

> ### Author Response · Authors · 2024-11-24
> **Response to More Visualizations, Key Difference from CamCo, Performance Improvement, Connection between Proposed New Insights about Noised Conditions, and Evaluation Benchmark Improvement**
>
> # Response to More Qualitative Results
> Please refer to **provided supplementary, and open the index.html**  for additional generated videos.
>
> # Response to Key Difference between CamCo and Ours
> We highly acknowledged the contribution of CamCo(arxiv in June) and have correctly cited it in our first submission. We will follow your adivces to further emphasize the difference/improvement between CamCo and our CamI2V. In addition, we will also add citation of CVD and more description about it in our paper.
>
> Please refer to **the general response: "To all reviewers: 3. Key Differences Between CamCo and CamI2V (Ours)"**
>
> Please refer to the **table** in **general response: "To all reviewers: 1.The Connection Between Our Insights on Conditions and the Proposed Method"**, where we follow your suggestion to add the quantative results of **epipolar attention only on reference frame (similar to CamCo**, but camco implements by resampling visible points while we implement by attention mask).
>
> # Response to  Performance Improvement
>
> Our method demonstrates significant improvements over CameraCtrl, achieving a **32.96% reduction in Rotation Error**, a **25.64% decrease in CamMC**, and a **20.77% improvement in Translation Error**, **without decrease in FVD**. These results were obtained using Text and Image CFG set to 7.5, 25 steps, and **camera CFG set to 1.0 (no camera cfg)**.
>
> Compared with CamCO-like (arxiv in June) method, we improve **17.08%, 6.61%, 9.00%** on RotErr, TransErr, and CamMC without FVD decrease, respectively.
>
> # Response to Proposed New Insights about Noised Conditions
> Other reviewers have highly acknowledged our insights on clean and noised conditions. We have made substantial efforts to present these new perspectives to share this understanding with the research community. We apologize for not effectively connecting these insights with our proposed method in the initial submission.
>
> Please refer to **the general response to all reviewers: 1. The Connection Between Our Insights on Conditions and the Proposed Method** for a detailed explanation.
>
> # Response to Evaluation Benchmark Improvement
> We have identified that current evaluation benchmarks are unreliable, and no open-source evaluation code is available. To address this, we propose several improvements:
> * Performing multiple measurements (averaging over five runs) to ensure consistency.
> * Conducting extensive evaluations on a large number of samples (1000 samples) for statistical significance.
> * Replacing the SfM algorithm with a more robust GLOMAP (**new work of COLMAP's author, small improvement**) to improve accuracy.
> * Computing relative poses with respect to the initial frame to standardize evaluations.
> * Introducing normalization to address the issue of unknown global scene scale caused by SfM predictions (**the most important improvement**).
> * Providing SFM with Ground-truth camera intrinsics.
>
> **The most important improvement is normalization to address the issue of unknown global scene scale**. It's described in RealEstate10K's paper that **"there is no way to determine global scene scale, so our reconstructed camera poses are up to an arbitrary scale per clip"**. Although they attempt to scale normalize the scene scale according to the 5% near depth planes, the camera pose for camera-controlled image-to-video generation is slightly different for different scenes and thus remains relative scene scale. Therefore, it's necessary to calculate normalized (relative) metrics if we can not control the global scene scale.
>
> We also commit to open-sourcing all our training and evaluation code, as well as the checkpoints. Although these modifications are small and straightforward, they hold significant importance for improving reliability and reproducibility in the community.
>
> # Final
>
> We sincerely thank you for your time and valuable feedback on our paper. Your insights have been instrumental in improving our work. We have carefully addressed the concerns raised and hope that our rebuttal provides a clearer understanding of our contributions.
> We wholeheartedly wish that your own submissions receive positive feedback and successful outcomes. We kindly ask that you take the time to thoroughly review our rebuttal.
>
> Regardless of the final decision, we remain committed to open-sourcing all our code and checkpoints to support the community's growth and development. Thank you once again for your consideration and support.

---

> ### Comment · Reviewer_yhFg · 2024-11-28
>
> I appreciate the authors' thorough response and additional experiments addressing my previous concerns. After reviewing the rebuttal and other reviews, I have several remaining concerns about this work's technical novelty.
>
> My primary concern relates to the paper's core technical contribution. Epipolar-based attention has been widely adopted in previous image/video generation works ([1], [2], [3], [4]), with [4] specifically implementing epipolar attention on noisy frames. While the authors present the register token as their main contribution, this appears incremental - one could achieve similar results by applying attention to all pixels (or sub-sampled pixels) when the epipolar line falls outside the frame. The authors note that their reproduction of CamCo results in NaN without the register token, which suggests CamCo may already employ similar techniques. The major benefit of register token as I understand is that it can reduce the memory cost to be O(HWL), yet the method still uses attention masks in their implementation which theoretically is still O(HW^2).
>
> However, I do acknowledge two valuable contributions:
>
> This appears to be the first model applying epipolar attention across all frames simultaneously, rather than the one-to-one approach of previous works. Experiments demonstrate this improves camera accuracy.
> The proposed Glomap-based evaluation could largely benefit the research community if made publicly available.
>
> Overall, while this work makes some improvements to epipolar line-based camera-controlled video diffusion, its technical insights remain limited to me. Therefore, I recommend a borderline reject.
>
> References:
> [1] Tseng et al., "Consistent View Synthesis with Pose-Guided Diffusion Model"
> [2] Huang et al., "EpiDiff: Enhancing Multi-View Synthesis via Localized Epipolar-Constrained Diffusion"
> [3] Xu et al., "CamCo: Camera-Controllable 3D-Consistent Image-to-Video Generation"
> [4] Kuang et al., "Collaborative Video Diffusion: Consistent Multi-video Generation with Camera Control"

---

> > ### Author Response · Authors · 2024-11-28
> > **About Contribution**
> >
> > Thank you sincerely for taking the time to provide detailed feedback. We deeply appreciate your support in raising your score from 3 to 5. However, we respectfully disagree with the relationship between technical part of contribution and the overall contribution.
> >
> > # About Core Contribution of this paper
> > **The core contribution of our paper is not the technical implementation of epipolar attention** but rather **identifying a critical challenge** in camera-controlled image-to-video generation: **the need for better modeling of cross-frame interactions, particularly under noisy settings**, to enhance geometric consistency and camera controllability. Highlighting such a fundamental issue is, in our view, a major contribution. By identifying this challenge, we open the door for the community to develop more effective solutions. **Epipolar attention is simply one of our initial attempts to address this problem**.
> >
> > Moreover, we introduce **a novel perspective that has been recognized by other reviewers**—the concept of **clean and noisy conditions—and how the effectiveness of a condition is inherently linked to the amount of uncertainty it can reduce**. It can also be seen from experiments that Dynamicrafter without camera-condition achieves much worser FVD than methods conditioned with camera pose. This means camera condition also help improving generation quality instead of only introducing camera control. This is considered the contribution of more condition. We believe this perspective offers valuable insights for future research, **helping others understand how to reduce uncertainty while modeling noisy cross-frame interactions**.
> >
> > Simple implementation is not inherently a weakness. Often, the **motivation**, **findings**, and **sharing of innovative and thought-provoking ideas** are the most significant contributions a paper can make. For instance, recaptioning in implementation part only adds more annotations on text caption. DIT’s major implementation change is replacing U-Net with a transformer, and SORA shifts from 2D VAE to 3D VAE along with DiT, which were also proposed in previous methods. Register tokens, 2024 ICLR Outstanding Paper Awards, only added several tokens in implementation attention.  These works demonstrate that **contributions can stem from straightforward implementations grounded in strong motivations and insights**.
> >
> > Similarly, **while our use of epipolar attention is straightforward, it serves as a proof of concept to validate our broader motivation and findings**. We hope this clarifies the value of our contributions.
> >
> > # About core contribution in technical/implementation part
> > We acknowledge that the register token is a relatively minor addition and could, as you suggested, be replaced with a less efficient full attention approach. Its primary purpose is indeed as a safeguard to ensure model stability and prevent NaN issues. The main contribution in this part is that we raise reasearchers' attention on situations of no epipolar lines, due to the common large movement and pure rotation in the practical application of camera-controlled image-to-video. This is different from 3D generation where object is centered, background is uniformly black/gray/white, scene is static, and camera is rotation around. In addition, epipolar attention, implemented via attention masks of memory-efficient attention, is actually more efficient in practice, despite large theoretical complexity.
> >
> > Our most significant technical contribution lies in proposing epipolar attention **across all frames**, which focuses on modeling **noisy cross-frame interactions**. As you noted, this may be one of the first implementations of this approach, although we refrain from claiming to be the very first since there are a few works that also apply attention across all frames. However, these prior works may not explicitly show the design principles underlying their approach. In contrast, our method is firmly rooted in the concept of **noisy cross-frame interactions**, providing a clear rationale for applying epipolar attention to all frames and setting our work apart from existing methods.
> >
> > **I am indeed not willing to only over-emphasize the performance improvement over sota** methods (CameraCtrl), despite 32.96%, 25.64%, 20.77% improvement in RotErr, CamMC, TranErr without decrease in FVD and 17.08%, 6.61%, 9.00%  improvment over CamCo-like methods (arxiv). **Sharing novel insights with community** remains my best wishes and we really need your support now.
> >
> > We hope that our response has clarified our contributions and the unique perspective our work brings to the field. If you have any further concerns or questions, please do not hesitate to reach out. We look forward to your response and will remain available to address any additional feedback.

---

### Official Review · Reviewer_D8XB · 2024-11-04

**Soundness:** 3
**Presentation:** 2
**Contribution:** 2
**Rating:** 6
**Confidence:** 4

**Summary:**

The paper proposes a new method to inject camera pose conditions for controlled video generation using a diffusion model. The method uses Plücker coordinates to represent the camera rays and introduces epi-polar mask attention for cross-view attention, which is more efficient than the full 3D attention counterpart. The proposed method is evaluated on the RealEstate10K dataset and achieves state-of-the-art performance.

**Strengths:**

+ The epi-polar mask attention layer proposed in the paper helps to enhance the camera control ability for video diffusion models. Also, it is plug-and-play – we don't need to retrain other modules of the original pretrained VDM.

+ It is an interesting idea to include the register token to handle cases where the epipolar constraint on correspondences fails, although more discussion and evaluation on it are needed (see the weakness section).

+ The paper addresses the inaccuracy in the SfM for robust evaluation metrics. Specifically, GLOMAP, a more state-of-the-art dense SfM pipeline, is used to validate the camera pose consistency.

**Weaknesses:**

+ Discussion and experiment missing for a key statement in the paper: While the paper states (in L112) that register tokens are included to handle rapid camera movements, occlusions, and dynamic objects, this contribution (also the key difference from CamCo) is not discussed in more detail. For example, how does this additional token help to deal with the non-epipolar constrained correspondences? Can the image-level register token handle pixel-level dense correspondences across frames (like moving arms of people)? In addition, the register token is not ablated, so it’s unclear if it actually helps for dynamic scene generation.

Furthermore, since this is the key difference from CamCo, more discussion and evaluation could help to distinguish this submission from CamCo.

+ Some parts of the presentation are confusing: While the discussion in Fig. 1 and Sec. 1 on different types of conditions based on how much uncertainty the condition can reduce is interesting, how is that related to the epipolar condition? Does the epipolar condition reduce more uncertainty, hence making it a good condition? How does this view of condition serve as a principle to introduce the epipolar condition in the paper? More clarification is needed.

+ For the quantitative results, cross-view consistency is missing, although it has been highlighted in the qualitative subsection (Fig. 7). Is there any reason why it is ignored in the tables as a metric? Cross-frame consistency can be another useful metric to show how the epipolar attention helps, in addition to camera controllability.

+ In Fig. 5, the multi-resolution epipolar mask is shown but not discussed in the text. What is the motivation for using a multi-resolution epipolar mask?

**Questions:**

Please refer to the weakness section above.

---

> ### Author Response · Authors · 2024-11-24
> **Response to Noisy Conditions, More Qualitative Results, Cross-View Consistency, Key Difference between CamCo and Ours**
>
> # Response to More Qualitative Results
> Please refer to **provided supplementary, and open the index.html.**
>
> # Response to Q1: The Connection Between the Proposed Noisy Condition and Epipolar Attention
>
> **Please also refer to our general response titled “1. The Connection Between Our Insights on Conditions and the Proposed Method” for additional context.**
>
> Our motivation is to understand **how noise affects cross-frame interactions** and to design strategies that mitigate the misleading effects of noise. In clean feature spaces, issues like feature copying due to large receptive fields can be mitigated by increasing model capacity and improving feature extraction. The main challenge in such settings is incorrect attention allocation when the aggregation space is too large.
>
> However, **in noisy feature spaces, the situation is different**. Due to the added noise, dissimilar features may erroneously receive high attention scores. In this scenario, even powerful models struggle to reconstruct clean pixels from random noise—especially at high noise levels—because the noise can mislead the attention mechanism, resulting in degraded performance.
>
> Our novel insight is that **simply increasing the receptive field, such as using 3D full attention, leads to a greater amount of noisy conditions.** Due to the randomness inherent in the noise, **accessing more noisy conditions does not necessarily reduce uncertainty and thus not necessarily lead to better performance**. This understanding suggests that models should focus on aggregating information from more reliable locations to effectively reduce uncertainty caused by noise. The idea of **uncertainty**  introduced by noisy features but not clean features leads to a new insight.
>
> **We have found that applying epipolar constraints is one of the most suitable way to prevent the model from being misled by noise.** By restricting attention to features along the epipolar lines, the model can interact with more relevant and **less noisy information**, improving cross-frame interactions in diffusion models. We believe these new insights are significant as they may inspire researchers to develop new architectural designs that better handle noisy conditions in generative models.
>
>
> # Response to Cross-View Consistency
>
> We appreciate the suggestion to evaluate cross-view consistency, recognizing it as a valuable metric for assessing our model's performance. Currently, we **follow MotionCtrl to calculate CamMC, CameraCtrl to determine Rotation Error (RotError) and Translation Error (TransError), and CamCo to transfer relative camera poses**. However, **these methods do not directly measure cross-view consistency**. While we acknowledge the importance of cross-view consistency, we are uncertain about the most appropriate metrics to apply. We would greatly appreciate **any recommendations on suitable metrics**.
>
> In our current evaluation, cross-view consistency is implicitly reflected in the qualitative results of the generated videos. For quantitative assessments such as CamMC, RotError, and TransError, we employ an **SFM-based approach to identify similar key points** and perform bundle adjustment to optimize camera poses. **This process inherently involves a form of cross-view consistency by ensuring that key points are consistently tracked across frames**. Consequently, **improved camera controllability often correlates with enhanced cross-view consistency**.
>
> Our method demonstrates significant improvements over CameraCtrl, achieving a **32.96% reduction in Rotation Error**, a **25.64% decrease in CamMC**, and a **20.77% improvement in Translation Error**, **without any decrease in FVD**. These results were obtained using Text and Image CFG set to 7.5, 25 steps, and **camera CFG set to 1.0 (no camera cfg)**.
>
> # Response to Key Difference between CamCo and Ours
> Please refer to **the general response: "To all reviewers: 3. Key Differences Between CamCo and CamI2V (Ours)"**

---

> ### Author Response · Authors · 2024-11-24
> **Response to Multi Resolution Mask and Image-Level Register Token**
>
> # Response to Multi Resolution Epipolar Mask
> Dynamicrafter employs a **UNet-based** diffusion model with multiple attention resolutions, such as **32×32, 16×16, 8×8, and 4×4 for input images at a resolution of 256×256**. When integrating epipolar attention as a plug-and-play module, we freeze the original spatial and temporal attention mechanisms and add extra epipolar attention at these various latent attention resolutions.
>
> Additionally, to efficiently train models at higher resolutions (512×320 and 1024×576), we **choose not to add epipolar attention at certain latent resolutions**. Specifically, for the 512×320 model, **epipolar attention is replaced by 3d full attention (extremely memory-efficient
>  due to optimiation of flash-attn)** at the 32×32 resolution, and 32×32 and 16×16 resolutions for 1024×576 model. This selective integration allows for minimal memory increase, enabling us to perform inference on the 1024×576 model with only 18 GB of memory, little memory increase compared to dynamicrafter baseline.
>
> This flexible approach ensures that our method remains memory-efficient while maintaining high performance across different image resolutions.
>
> # Response to Register Token
>
> Please refer to **the general response "to all reviewers: 4.Regarding the Register Token."**
>
> When **epipolar line exists but is far away from the queried frame**, the zero length kv tokens in memory-efficient attention **without register tokens leads to NaN during training, making it impossible to train a model**. In this case, two added register tokens serve as **placeholder to ensure the non-zero length of kv tokens**.
>
> Despite utilizing only two register tokens is insufficient, it **ensures the model's basic functionality and only part of pixels on the current frame who have no epipolar line degrade to CameraCtrl without epipolar attention** (it can also be considered as a kind of epipolar attention as in this situation, it is appropriate to aggregate no features under epipolar constraint).
>
> Now we apply the token-level register token (only two tokens are added like cls token) but image-level register token can also be applied. For example, in image-to-video generation, the pixels of the reference image can always be seen and serve as image-level register tokens. However, this may also lead to dynamics decrease and it can be seen from the results of 3D full attention (too many noisy conditions) and epipolar only on reference frame (dynamics decrease due to always copying from reference frame, seen in videos of supplementary).

---

### Official Review · Reviewer_L7Wx · 2024-11-04

**Soundness:** 3
**Presentation:** 2
**Contribution:** 2
**Rating:** 6
**Confidence:** 5

**Summary:**

This paper extends an existing I2V method (DynamiCrafter) to accept additional camera viewpoint control. The input cameras are parameterized as Plücker rays, serving as a position encoding. The authors additionally inset learnable Epipolar attention layers before temporal attn, which explicitly model cross-view geometry to further enhance the adherence to the input camera.

**Strengths:**

The paper is well written with a few nicely created figures, e.g., Fig. 1 and Fig. 2. The ideas of (1) clean vs. noisy condition & (2) register tokens are neat (but they are also related to the weakness and questions below). Despite tuned with static/rigid scene dataset -- RealEstate10k, from the few generated videos in the supplementary, the motion dynamic of foreground is not lost too much. The discussion in L457-479 is good and clearly supports the design choices. For fair comparison, the authors spend effort reproducing the recipes of baselines (MotionCtrl and CameraCtrl) in their framework (instead of applying the released ckpt out of the box) such that they all stand on a common ground.

**Weaknesses:**

* Contribution is clearly articulated in L.135-139 but not verified in the experiment: using Plücker embedding and/or epipolar attention has become a norm in the 3D generation literature, e.g. [1]. Applying one of them, if not both, in the video generation field has also been done, e.g., CameraCtrl, CamCo etc. Therefore, the biggest technical contribution I see in this work, to my knowledge, is applying the idea of register tokens to account for occlusion, zero epipolar scenarios, etc. Despite simple, I find this idea neat, but I don't see any experiments ablating this key idea to analyze the effect. A contribution/novelty has be verified by the experiments. If this concern can be addressed, I am happy to raise the rating.

* Method description is a bit insufficient. I need to read Sec. 5.3 to realize Plücker rays are also used as global positional encoding similar to CameraCtrl. This is also part of the final method so it has to be described in Section 3.



[1] Kant et al., SPAD : Spatially Aware Multiview Diffusers, CVPR24. https://yashkant.github.io/spad/

**Questions:**

1. The concept of clean vs. noisy conditions is also neat. The authors even make a figure to illustrate it (Fig. 1), but the knowledge of clean vs. noisy conditions seems not fully exploited in the method? For example, Fig. 1 points out that text and RT are clean conditions, but I don't see this new insight leads to new architectural designs? Adding camera information through Plücker embedding and epipolar attention is a natural choice due to the nature of multi-view geometry, not because it is a clean condition. I wonder if making such separation is really necessary in the exposition.  (This is more of a question for presentation, not technical details.)

2. More generated video results. All video examples in the supplementary material show only zooming-out camera motion, which, in my experience, is the easiest one to be learnt by the network. Please provide other examples, such as panning left/right, moving up/down, etc.

---

> ### Author Response · Authors · 2024-11-24
> **Response to more visualizations, the necessity of new insights, and zero epipolar situations.**
>
> ## Response to Q2: More Qualitative Results
> **Please refer to provided supplementary, and open the index.html.**
>
>
> ## Response to Q1: The Connection Between the Proposed Noisy Condition and Epipolar Attention, and the Necessity of These New Insights
>
> We appreciate your question regarding the relationship between our concept of noisy conditions and epipolar attention.
> **Please also refer to our general response titled “1. The Connection Between Our Insights on Conditions and the Proposed Method” for additional context.**
>
> Previous methods have introduced epipolar attention primarily with the motivation of enhancing geometric consistency, **but these approaches were not specifically designed for settings with inherent noise**. In contrast, our work focuses on the unique challenges posed by the noisy feature spaces inherent in diffusion models.
>
> Our motivation is to understand **how noise affects cross-frame interactions** and to design strategies that mitigate the misleading effects of noise. In clean feature spaces, issues like feature copying due to large receptive fields can be mitigated by increasing model capacity and improving feature extraction. The main challenge in such settings is incorrect attention allocation when the aggregation space is too large.
>
> However, **in noisy feature spaces, the situation is different**. Due to the added noise, dissimilar features may erroneously receive high attention scores. In this scenario, even powerful models struggle to reconstruct clean pixels from random noise—especially at high noise levels—because the noise can mislead the attention mechanism, resulting in degraded performance.
>
> Our novel insight is that **simply increasing the receptive field, such as using 3D full attention, leads to a greater amount of noisy conditions.** Due to the randomness inherent in the noise, **accessing more noisy conditions does not necessarily reduce uncertainty and thus not necessarily lead to better performance**. This understanding suggests that models should focus on aggregating information from more reliable locations to effectively reduce uncertainty caused by noise. The idea of **uncertainty**  introduced by noisy features but not clean features leads to a new insight.
>
> **We have found that applying epipolar constraints is one of the most suitable way to prevent the model from being misled by noise.** By restricting attention to features along the epipolar lines, the model can interact with more relevant and **less noisy information**, improving cross-frame interactions in diffusion models. We believe these new insights are significant as they may inspire researchers to develop new architectural designs that better handle noisy conditions in generative models.
>
> ## Response to Q1: design for epipolar attention when epipolar lines vanish
>
>
> Please also refer to our general response titled **4. Regarding the Register Token** for additional context.
>
> We apologize for not clearly presenting the details regarding this issue earlier.
> When we attempted to **apply epipolar attention without register tokens, we encountered NaN** errors during training and **were unable to successfully train a model for ablation studies**.
>
> Our novelty lies in addressing situations where epipolar lines vanish, which is crucial for robust performance in practical applications. Regarding this scenario, we have observed that previous works on epipolar attention often do not provide special designs or detailed descriptions for handling cases where epipolar lines are absent. This gap motivated us to develop an effective solution.

---

> > ### Comment · Reviewer_L7Wx · 2024-12-02
> >
> > I've read the authors' all response thoroughly and thank authors for preparing good ablation and new qualitative results.
> > As all my concerns are addressed I will raise my rating to border accept.

---

### Author Response · Authors · 2024-11-23
**To all reviewers: The Connection Between Our Insights on Conditions and the Proposed Method**

We sincerely thank all reviewers and PCs for their valuable feedback and time.
We are gratified that the reviewers appreciated the interest of our new insights regarding clean and noisy conditions,
and how much the uncertainty reduced by a condition determines its quality.
This affirms that the considerable effort we invested in presenting this viewpoint at the beginning of the paper was worthwhile.
We apologize for not effectively connecting it with our proposed method.
We will now clearly elucidate the connection between noisy condition and our proposed method.
In addition, it’s strongly recommended to **check the visualizations in supplementary/index.html**.

# 1. The Connection Between Our Insights on Conditions and the Proposed Method
**Adding more conditions to generative models typically reduces uncertainty and improves generation quality**
(e.g. providing detailed text conditions through recaption).
In this paper, we argue that it is also crucial to consider **noisy conditions** like latent features $z_t$, which contain valuable information along with random noise.
For instance, in SDEdit for image-to-image translation, random noise is added to the input $z_0$ to produce a noisy $z_t$.
The clean component  $z_0$  preserves overall similarity, while the introduced noise leads to uncertainty, enabling diverse and varied generations. In this paper, we argue that **providing the model with more noisy conditions, especially at high noise levels, does not necessarily reduce more uncertainty, as the noise also introduces randomness and misleading**. This is the key insight we aim to convey.

To validate this point, we designed experiment with the following setups:

* **Plücker Embedding (Baseline, CameraCtrl)**: This setup has minimal noisy conditions on cross frames due to the inefficiency of the indirect cross-frame interaction (spatial+temporal attention).

* **Plücker Embedding + Epipolar Attention Only on Reference Frame**: Similar to **CamCo**, this setup treats the reference frame as the source view, enabling the target frame to refer to it. It accesses **a small amount** of noisy conditions on the reference frame. However, some pixels of the current frame may have no epipolar line interacted with reference frame, causing it to degenerate to a CameraCtrl-like model without epipolar attention.

* **Plücker Embedding + Epipolar Attention (Our CamI2V)**: This setup can impose epipolar constraints with all frames (including adjacent frames that have interactions in most cases to ensure an sufficient amount of noisy conditions).

*  **Plücker Embedding + 3D Full Attention**: This setup allows the model to directly interact with features of all other frames, accessing the most noisy conditions.

**The amount of accessible noisy conditions of the above four setups increase progressively**.  One might expect that 3D full attention, which accesses the most noisy conditions, would achieve the best performance.  However, as shown in the following **table**,  3D full attention performs only slightly better than CameraCtrl and is inferior to CamCo-like setup. Notably, our method achieves best result by interacting with more noisy conditions along the epipolar lines. It can also be clearly seen in **comparison in supplementary's index.html** that camco-like setup reference much on the first frame and cannot generate new objects. The 3D full attention generates objects within large movement due to its access to all frames pixels while it is affected by incorrect position of pixels.  These findings confirm our insight that **an optimal amount of noisy conditions leads to better uncertainty reduction, rather than merely increasing the quantity of noisy conditions.**

| Method                  | RotErr$\downarrow$ | TransErr$\downarrow$ | CamMC$\downarrow$ | FVD (VideoGPT)$\downarrow$ | FVD (StyleGAN)$\downarrow$ |
|-----------------|---------------|---------------|---------|---------|---------|
| DynamiCrafter            | 3.3415             | 9.8024       | 11.625   | 106.02      | 92.196     |
| +MotionCtrl          | 0.8636             | 2.5068         | 2.9536            | 70.820               | 60.363     |
| +Plucker Embedding (Baseline, CameraCtrl)    | 0.7098             | 1.8877               | 2.2557     | **66.077**        | 55.889      |
| +Plucker Embedding + Epipolar Attention Only on Reference Frame (CamCo-like) | 0.5738    | 1.6014               | 1.8851            | 66.439     | 56.778      |
| +Plucker Embedding + Epipolar Attention (Our CamI2V)    | **0.4758**         | **1.4955**      | **1.7153**        | 66.090       | **55.701**     |
| +Plucker Embedding + 3D Full Attention   | 0.6299             | 1.8215   | 2.1315     | 71.026  | 60.00     |

---

> ### Author Response · Authors · 2024-11-25
> **Response to Different Quantative Value from the Initial Submission.**
>
> To quickly finish the experiments, we change to six machines equipped with eight H100 GPUs and reproduce all the experiments. We also train from scratch 512x320 model for generating visualizations in index.html. All experiments are conducted on the same baseline with same seed by changing only a few settings in config file to ensure a fair comparison. The results are tested with the same seed on the same 1k samples of test set.
>
> The primary factor contributing to performance differences was that, in the initial draft, our CameraCtrl and MotionCtrl models employed a fixed condition frame index, consistently using the first frame as the reference image. In contrast, our CamI2V model is designed to support any frame as the reference by using a random condition frame index during training.
>
> To ensure a fair and transparent comparison between the methods, **the latest version of our submission uniformly employs a random condition frame index across all models**. Additionally, to mitigate the influence of camera CFG, **we set the camera CFG to 1.0 (no camera CFG) while in our first submission we use camera cfg 1.875**, where an overemphasis on camera controllability may result in a substantial decrease in generation quality.
>
> We believe these changes provide a more equitable evaluation of our proposed method and **clearly highlight the differences between the approaches.**

---

### Author Response · Authors · 2024-11-23
**To All Reviewers: Key Differences Between CamCo and Ours**

## 2. The motivation of this paper
With the extreme optimization of memory-efficient attention techniques, simply employing 3D full attention has already demonstrated remarkable 3D consistency in generated videos (e.g., OpenSora, CogVideoX). **This raises the question of whether merely scaling up models can address all challenges, while structural modifications might inadvertently degrade performance.**
Motivated by this, we **aimed to determine whether epipolar attention is effective in practice, especially in models under noisy settings like diffusion models, as opposed to its use in autoregressive models and pure transformers**. Our experiments support our novel insights on the role of noisy conditions in cross-frame interaction. Furthermore, we hope that sharing these new insights will provide readers with a more inspiring understanding and, by open-sourcing our work, enable more researchers to advance the development of this field.

## 3. Key Differences Between CamCo and CamI2V (Ours)

Firstly, we have great respect for the work of CamCo and have properly cited their contributions in our submission. While we acknowledge their significant advancements in the field, we would like to highlight several key differences between our approach and CamCo:

* **Application of Epipolar Attention Across Frames**: CamCo applies epipolar attention only to the reference frame (specifically, the first frame). In contrast, our method applies epipolar attention across all frames, ensuring a sufficient amount of noisy conditions and allowing the reference frame to be at an arbitrary position. This difference may stem from CamCo’s motivation, which originates from the concepts of “source view” and “target view” in the fields of 3D and multi-view generation to enhance geometric consistency. Our motivation, however, is based on the concept of \textit{noisy conditions**, leading to a different approach in handling cross-frame interactions.

* **Handling Missing Epipolar Intersections**: To the best of our knowledge, previous methods do not provide detailed explanations on how they handle situations where some pixels in the current frame have no intersecting epipolar lines in the reference frame. This issue arises not only in scenarios with large movements and pure rotations but also in common settings such as zooming out, panning, and rotating, where edge pixels disappear or new pixels appear. In 3D generation tasks, this problem is less pronounced due to simple camera movements where objects are typically centered, backgrounds are uniform, scenes are static, and the camera rotates around the object. In practical applications, the absence of epipolar lines is significant because it can lead to computational issues, such as NaN values resulting from zero-length key/value vectors in attention mechanisms. We address this problem by introducing a **register token**, which we discuss in detail later.

* **Implementation Differences**: CamCo proposes constructing epipolar lines by resampling visible pixel locations, avoiding variable-length epipolar line features, and improving the theoretical complexity from  $O((hw)^2)$  to  $O(hwl)$. However, we were unable to efficiently reproduce this approach and found it incompatible with current memory-efficient attention mechanisms, resulting in significantly increased memory usage and reduced training speed. This may be due to limitations in our reproduction, and CamCo’s code was not available. In our method, we implement epipolar attention using attention masks optimized for memory-efficient attention techniques, resulting in minimal increases in memory usage and computation time.

Our model with epipolar attention can be trained on GPUs with 24GB of memory and requires less than 12GB for inference at resolutions of 256×256 and 512×320. For a resolution of 1024×576, approximately 30GB and 18GB are required for training and inference, respectively. Detailed running times and memory usage are listed in the appendix. We emphasize the importance of training and inference efficiency for practical use, which is why we dedicate significant space in the method section to discuss our implementation of epipolar attention.

---

### Author Response · Authors · 2024-11-23
**To All reviewers: Regarding the Register Token**

## 4. Regarding the Register Token

Our contribution lies on the **consideration of situations without epipolar lines**. Our core design idea is to **avoid distinctions in implementation** to maximize the acceleration provided by memory-efficient attention.
Therefore, we propose that **epipolar attention should still have placeholder tokens to query even when there are no tokens available.
This led us to introduce the register token**.

**We did not perform ablation studies on this component** because, **without the register token, directly applying epipolar attention based on attention masks can cause issues**. Specifically, when a pixel’s attention mask is all zeros, PyTorch’s memory-efficient attention implementation adds negative infinity to all entries, leading to **zero length of key/value tokens**. This results in **NaN** values during training, **making it impossible to train the model**. We apologize for not detailing this in the original manuscript. We are also unsure how other methods handle this situation, but it is quite common in camera-controlled image-to-video tasks, where pure rotations and large movements are common.

| Method                  | RotErr$\downarrow$ | TransErr$\downarrow$ | CamMC$\downarrow$ | FVD (VideoGPT)$\downarrow$ | FVD (StyleGAN)$\downarrow$ |
|-----------------|---------------|---------------|---------|---------|---------|
| DynamiCrafter            | 3.3415             | 9.8024       | 11.625   | 106.02      | 92.196     |
| +MotionCtrl          | 0.8636             | 2.5068         | 2.9536            | 70.820               | 60.363     |
| +Plucker Embedding (Baseline, CameraCtrl)    | 0.7098             | 1.8877               | 2.2557     | **66.077**        | 55.889      |
| +Plucker Embedding + register token + Epipolar Attention Only on Reference Frame (CamCo-like) | 0.5738    | 1.6014               | 1.8851            | 66.439     | 56.778      |
| +Plucker Embedding + register token + Epipolar Attention (Our CamI2V)    | **0.4758**         | **1.4955**      | **1.7153**        | 66.090       | **55.701**     |
| +Plucker Embedding + **no register token** + Epipolar Attention  | NaN         | NaN     | NaN       | NaN       | NaN    |

We categorize two cases where epipolar line do not exist:

* **Degeneration into Epipolar Planes**: During pure rotation, the epipolar line degenerates into an epipolar plane. In this scenario, the pixel does not disappear but rotates to another position in the image. By searching all pixels, we can locate the pre-rotation pixel, effectively making the attention mask all ones. In this case, the epipolar line is defined as $ax+by=0, a=0, b=0$ and becomes a plane, thus the distance from each pixel to this plane becomes zero. All the zero distance pass through our predefined threshold, making a full mask. This situation then becomes equivalent to full attention on that frame.

* **Epipolar Lines Falling Outside the Image**: Large movements can cause the epipolar line to fall outside the image area—the line exists, has no intersections with the region of image and the distance is too distant. In this case, using only two register tokens avoids the zero attention issue and ensures the succesfull calculation on attention scores. Despite utilizing only two register tokens is insufficient, it ensures the model's basic functionality and only part of pixels on the current frame who have no epipolar line degrade to CameraCtrl without epipolar attention (it can also be considered as a kind of epipolar attention as in this situation, it is appropriate to aggregate no features under epipolar constraint).

**Initially, we believed that the core contribution of the paper was the concept of noisy conditions rather than epipolar attention, which has been applied multiple times before.** Therefore, we did not provide detailed explanations, which is our fault.

## 5. Citation Updates Based on Reviewers’ Feedback
We will carefully review our citations in light of the reviewers’ comments
and update the Related Work section accordingly. Specifically,
we will include the papers mentioned by the reviewers
that utilize Plücker embeddings and epipolar attention in the fields of 3D generation
and multi-view synthesis.
If there are any additional references we may have missed, please feel free to inform us.

## 6. Improvement On Camera Controllability without FVD Decrease
Our method demonstrates significant improvements over CameraCtrl, achieving a **32.96% reduction in Rotation Error**, a **25.64% decrease in CamMC**, and a **20.77% improvement in Translation Error**, **without decrease in FVD**. These results were obtained using Text and Image CFG set to 7.5, 25 steps, and **camera CFG set to 1.0 (no camera cfg)**.

Compared with CamCO-like (arxiv in June) method, we improve **17.08%, 6.61%, 9.00%** on RotErr, TransErr, and CamMC without FVD decrease, respectively.

---

### Author Response · Authors · 2024-12-02
**Final Reminder and Request for Further Feedback**

We sincerely thank all reviewers for their valuable feedback and constructive discussions throughout the review process. As a friendly reminder, **only one day remains for submitting your responses**. This notice is prompted by the fact that some reviewers have not yet submitted their rebuttal responses, and others may still have unresolved questions or concerns.

We have worked diligently to address the majority of reviewers' concerns regarding **the connection between the novel insights and the proposed methods**. Specifically, we have clarified key points such as **the relationship between a condition and its ability to reduce uncertainty**, and identified one of the critical challenges in the field of camera-controlled image-to-video generation: **modeling cross-frame interactions under noisy settings**. Additionally, we **introduced the perspective of noisy/clean conditions to explain why applying epipolar constraints across all frames is essential**.

**We look forward to your continued support and hope you will consider raising your scores** to reflect the improvements made and the contributions of our work. If you have any further questions or concerns, please feel free to reach out—we remain available to address them.

Thank you again for your time and thoughtful feedback.

---

### Meta-Review · Area_Chair_fmhb · 2024-12-19

**Metareview:**

This paper presents an image-to-video diffusion model that enables camera control. The input cameras are represented as Plücker ray embeddings. Epipolar attention is used to aggregate cross-frame features and enhance cross-view geometry. The paper introduces register tokens as placeholders to avoid NaN in cases without intersections between frames.

While the proposed method has some acknowledged merits, reviewers have expressed concerns about its novelty and view its contributions as incremental. The area chair has also reviewed the paper and shares similar reservations. Despite the authors' efforts to clarify key contributions in their rebuttal and revision, the concerns remain. Consequently, the paper received final ratings of 5, 5, 6, and 6, which indicates that it does not have enough support for acceptance.

**Additional Comments On Reviewer Discussion:**

During the rebuttal, the authors modified the focus of their contributions, which put more emphasis on the perspective of noisy/clean conditions. The following quotes are excerpted from the authors' responses:
> * The core contribution of our paper is not the technical implementation of epipolar attention but rather identifying a critical challenge in camera-controlled image-to-video generation: the need for better modeling of cross-frame interactions, particularly under noisy settings. ... Epipolar attention is simply one of our initial attempts to address this problem. ... This means camera condition also help improving generation quality instead of only introducing camera control. This is considered the contribution of more condition. We believe this perspective offers valuable insights for future research, helping others understand how to reduce uncertainty while modeling noisy cross-frame interactions.
> * Our most significant technical contribution lies in proposing epipolar attention across all frames, which focuses on modeling noisy cross-frame interactions. ... However, these prior works may not explicitly show the design principles underlying their approach. In contrast, our method is firmly rooted in the concept of noisy cross-frame interactions, providing a clear rationale for applying epipolar attention to all frames and setting our work apart from existing methods.

The reviewers still had some concerns about the contributions even after the rebuttal.
* While it is acknowledged that the proposed method has led to improvements in epipolar-based camera-controlled video diffusion, the technical insights provided are considered limited.

* Although the modeling of noisy cross-frame interactions is well-motivated, the current interpretation appears to be intuitive rather than sufficiently convincing. To strengthen the submission, more technical insights or findings are needed, including intermediate observations and informative comparisons. This issue cannot be addressed with minor revisions and clarifications; major restructurings are required.

Moreover, although the authors offer some explanations, further investigation into the differences between the quantitative results in the updated revision and those in the original submission is required.

---

### Decision · Program_Chairs · 2025-01-22

Reject